# Activity-based CRISPR scanning uncovers allostery in DNA methylation maintenance machinery

Kevin Chun-Ho Ngan[1,2], Samuel M Hoenig[1], Hui Si Kwok[1,2], Nicholas Z Lue[1,2], Pallavi M Gosavi[1,2], David A Tanner[1], Emma M Garcia[1,2], Ceejay Lee[1,2], Brian B Liau[1,2]*

[1]Department of Chemistry and Chemical Biology, Harvard University, Cambridge, United States; [2]Broad Institute of MIT and Harvard, Cambridge, United States

**Abstract** Allostery enables dynamic control of protein function. A paradigmatic example is the tightly orchestrated process of DNA methylation maintenance. Despite the fundamental importance of allosteric sites, their identification remains highly challenging. Here, we perform CRISPR scanning on the essential maintenance methylation machinery—DNMT1 and its partner UHRF1—with the activity-based inhibitor decitabine to uncover allosteric mechanisms regulating DNMT1. In contrast to non-covalent DNMT1 inhibition, activity-based selection implicates numerous regions outside the catalytic domain in DNMT1 function. Through computational analyses, we identify putative mutational hotspots in DNMT1 distal from the active site that encompass mutations spanning a multi-domain autoinhibitory interface and the uncharacterized BAH2 domain. We biochemically characterize these mutations as gain-of-function, exhibiting increased DNMT1 activity. Extrapolating our analysis to UHRF1, we discern putative gain-of-function mutations in multiple domains, including key residues across the autoinhibitory TTD–PBR interface. Collectively, our study highlights the utility of activity-based CRISPR scanning for nominating candidate allosteric sites, and more broadly, introduces new analytical tools that further refine the CRISPR scanning framework.

*For correspondence: liau@chemistry.harvard.edu

## Editor's evaluation

This manuscript describes the use of genome editing screens to identify mechanisms underlying resistance to the hypomethylating anti-cancer agent decitabine, an activity-based inhibitor of the DNA methyltransferase DNMT1. A specific focus is given to the development of tools and approaches to identify allosteric mechanisms of resistance that emerge, including those that appear to act as gain-of-function mutations in DNMT1 and its interacting partner UHRF1. These findings showcase the power of large-scale genome editing for uncovering novel resistance mechanisms and investigating protein allostery.

## Introduction

Allostery is a fundamental property that enables proteins to translate the effect of a stimulus at one site to modulate function at another distal site. Despite intense study, the identification of allosteric sites across diverse protein targets remains challenging and highly contextual. Unlike orthosteric sites, allosteric sites are often less conserved between related proteins and the principles governing their structural features and properties are not as well understood (*Garlick and Mapp, 2020*; *Nussinov et al., 2011*). Due to these challenges, fewer experimental and computational approaches exist to identify and characterize allosteric sites (*Lu et al., 2014*). Nevertheless, there have been significant

efforts to develop small molecule allosteric modulators, as the structural diversity of allosteric sites offers the potential for greater selectivity, lower toxicity, and fine-tuning of protein function compared to orthosteric ligands (*Garlick and Mapp, 2020*; *Nussinov et al., 2011*). Therefore, the development of new tools that enable the identification of allosteric mechanisms would further our understanding of protein regulation and facilitate drug discovery.

Leveraging pharmacological and genetic perturbations in tandem has been widely successful for target deconvolution and elucidating drug mechanism of action (*Schenone et al., 2013*). In particular, the identification of drug resistance-conferring mutations provides critical validation of on-target engagement and can often illuminate underlying biology (*Freedy and Liau, 2021*). Although many resistance mutations occur proximally to the drug-binding site, they can also arise at distal positions of a target protein and operate by perturbing allosteric mechanisms (*Azam et al., 2003*; *Ragland et al., 2014*; *Henes et al., 2019*). For example, resistance mutations to ABL1 inhibitors, including both orthosteric and allosteric inhibitors, consistently arise outside the drug-binding site and drive resistance by destabilizing the inactive conformation or otherwise neutralizing ABL1 autoinhibition (*Azam et al., 2003*; *Adrián et al., 2006*; *Sherbenou et al., 2010*; *Lee and Shah, 2017*; *Xie et al., 2020*). Such findings raise the prospect that identifying distal drug resistance mutations, either in the direct target or in interacting partners, can be exploited to systematically discover and characterize allosteric mechanisms.

Recently, we and others have used CRISPR–Cas9 tiling mutagenesis screens to uncover modes of small molecule action by identifying drug resistance mutations (*Neggers et al., 2018*; *Donovan et al., 2017*; *Vinyard et al., 2019*; *Gosavi et al., 2022*; *Kwok et al., 2022*). In our approach, termed CRISPR-suppressor scanning, Cas9 is used to systematically mutate a target protein with a pooled library of single-guide RNAs (sgRNAs) spanning its entire coding sequence (CDS) to generate large numbers of diverse protein variants in situ (*Ngan et al., 2022*). This surviving cellular pool is then treated with small molecule inhibitors to identify variants conferring drug resistance. Because resistance mutations can occur beyond the drug-binding site, we posited that such mutations might operate by disrupting interactions involved in allosteric regulation of protein function (*Vinyard et al., 2019*; *Gosavi et al., 2022*; *Kwok et al., 2022*). However, such distal mutations, which generally exhibit partial resistance phenotypes, are often overshadowed by the enrichment of drug-binding-disrupting mutations that confer complete rescue to the drug. Consequently, we considered whether the use of an activity-based inhibitor that closely resembles the target protein's native substrate might disfavor the formation of binding site mutations and could therefore be exploited to preferentially identify distal resistance mutations and potential allosteric mechanisms.

## Results

### Activity-based CRISPR scanning nominates distal allosteric sites of DNMT1 and UHRF1

Novel approaches to study allostery are particularly attractive for chromatin-modifying enzymes, whose activities are strictly regulated for proper gene expression. A paradigmatic example is the establishment and maintenance of DNA methylation by DNA methyltransferases (DNMTs). Canonically, DNMT1 performs maintenance methylation, ensuring faithful propagation of the methylation landscape (*Smith and Meissner, 2013*). Accordingly, DNMT1 activity is tightly controlled to prevent aberrant methylation (*Song et al., 2011*; *Takeshita et al., 2011*; *Song et al., 2012*; *Zhang et al., 2015*; *Ren et al., 2018*). While structural and biochemical studies have uncovered important mechanistic details underlying DNMT1 regulation, in vitro studies cannot recapitulate the complexities of the cellular environment and dependence on cofactors such as UHRF1 for DNA methylation in vivo (*Bestor and Ingram, 1983*; *Fatemi et al., 2001*; *Vilkaitis et al., 2005*; *Sharif et al., 2007*; *Bostick et al., 2007*). These discrepancies highlight outstanding gaps in our understanding of DNMT1 regulation and underscore the importance of investigating allosteric regulatory mechanisms within their endogenous context.

We reasoned that CRISPR-suppressor scanning of *DNMT1* and *UHRF1* with the activity-based inhibitor decitabine (DAC) could uncover mechanisms of DNMT1 allosteric regulation. DAC is a clinically approved DNA hypomethylating agent that acts through mechanism-based inhibition of DNMTs (*Christman, 2002*; *Jabbour et al., 2008*). DAC is a near-identical analog of deoxycytidine—differing

only by two atoms—that is incorporated into genomic DNA during replication. When DNMT1 methylates DAC on the nascent strand, DAC's unique structure prevents DNMT1 release, forming covalent DNMT1–DNA adducts that drive DNMT1 degradation and subsequent global DNA hypomethylation (*Figure 1a*, *Figure 1—figure supplement 1a, b*). At higher DAC doses, these adducts cause acute cytotoxicity independent of hypomethylation (*Tsai et al., 2012*). Consistent with this mechanism, previous work has shown that reducing DNMT1 protein levels alleviates DAC-induced cytotoxicity by decreasing DNMT1–DNA crosslinks (*Jüttermann et al., 1994*). Because maintenance methylation is an essential process, we reasoned that mutations in *DNMT1* arising in response to DAC treatment would be subject to the following constraints: (1) active site mutations disrupting DAC binding but not substrate recognition are highly unlikely due to its near-identical resemblance to DNMT1's native substrate, deoxycytidine; (2) loss-of-function (LOF) mutations alleviating adduct-induced cytotoxicity by reducing *DNMT1* copy number may incur fitness penalties from defects in maintenance methylation. Consequently, such LOF mutations may accompany hypermorphic gain-of-function (GOF) mutations that compensate for reduced DNMT1 protein levels. In light of these considerations, we hypothesized that DAC's mechanistic requirements could be exploited as an activity-based selection to preferentially enrich for distal GOF mutations that alter DNMT1 allostery and enhance its activity.

We performed CRISPR scanning in K562 cells using a pooled sgRNA library targeting the *DNMT1* and *UHRF1* coding sequences (*Figure 1b*). After lentiviral transduction of Cas9 and the sgRNA library, the cellular pool was split and treated with vehicle (dimethyl sulfoxide; DMSO) or DAC for 8 weeks. We collected genomic DNA from the surviving cells and performed targeted amplicon sequencing of the sgRNA cassette to quantify sgRNA frequencies. We then normalized sgRNA frequencies to their initial frequencies in the library and calculated 'resistance scores' by comparing relative sgRNA abundance in DAC versus vehicle treatment (*Figure 1c, d*, *Supplementary file 1*).

We observed the enrichment of numerous sgRNAs after prolonged DAC treatment, consistent with the emergence and expansion of drug-resistant populations. As expected, the majority of these enriched sgRNAs targeted *DNMT1* versus *UHRF1* (*Figure 1c, d*). Many top enriched *DNMT1* sgRNAs targeted N-terminal regulatory domains (e.g., RFTS, CXXC, and BAH2), supporting the notion that resistance mutations may arise in regions distal from the active site (*Figure 1c*). Indeed, the top enriched sgRNA in the screen, sgD702, targeted the CXXC–BAH1 linker region. Moreover, we also detected enriched sgRNAs targeting the UBL, TTD, and SRA domains of UHRF1, suggesting that mutations beyond the direct drug target may also confer a selective advantage to DAC (*Figure 1d*).

While our activity-based CRISPR scanning approach using DAC enriched for many sgRNAs targeting regions outside the catalytic domain of DNMT1, we have previously shown that reversible inhibitors can also select for sgRNAs targeting regions distal to their binding sites; however, the enrichment of such sgRNAs is generally overshadowed by those targeting the drug-binding site (*Vinyard et al., 2019*; *Gosavi et al., 2022*; *Kwok et al., 2022*). Therefore, to more accurately assess whether the use of an activity-based inhibitor such as DAC can predispose a CRISPR scanning experiment toward the enrichment of distal sgRNAs, we conducted CRISPR-suppressor scanning of *DNMT1* and *UHRF1* using the non-covalent, reversible DNMT1 inhibitor GSK3484862 (*Pappalardi et al., 2021*) (GSKi, *Figure 1—figure supplement 1d*) to enable a head-to-head comparison of sgRNA enrichment profiles between activity-based and reversible inhibitors (*Figure 1e, f*). Whereas the top enriched sgRNA under DAC treatment targeted DNMT1 residue D702 in the CXXC–BAH1 linker (*Figure 1c*), the top enriched sgRNA under GSKi treatment targeted DNMT1 residue H1507, located in the target recognition domain (TRD), a subregion of the DNMT1 catalytic domain (*Figure 1e, h*, *Figure 1—figure supplement 1e*). Notably, previous work has demonstrated that H1507 directly interacts with and contributes to the binding of the structurally related GSKi derivatives GSK3685032 and GSK3830052 (*Figure 1h*, *Figure 1—figure supplement 1e*), with the H1507Y mutation reducing GSK3685032 inhibition of DNMT1 by >350-fold compared to wild-type DNMT1 (*Pappalardi et al., 2021*). Further comparison of sgRNA resistance scores across DAC and GSKi treatment conditions revealed highly distinct sgRNA enrichment profiles, with no sgRNAs labeled as hits in both conditions (*Figure 1g*). Taken together, our data support the notion that activity-based and reversible inhibitors may exert differential selective pressures that lead to unique enrichment profiles.

As sgRNAs enriched under DAC treatment targeted diverse regions spanning *DNMT1*, we next considered whether their enrichment might indicate mutational hotspots. First, we investigated whether *DNMT1*-targeting sgRNAs exhibited any linear clustering, defined as contiguous amino acid

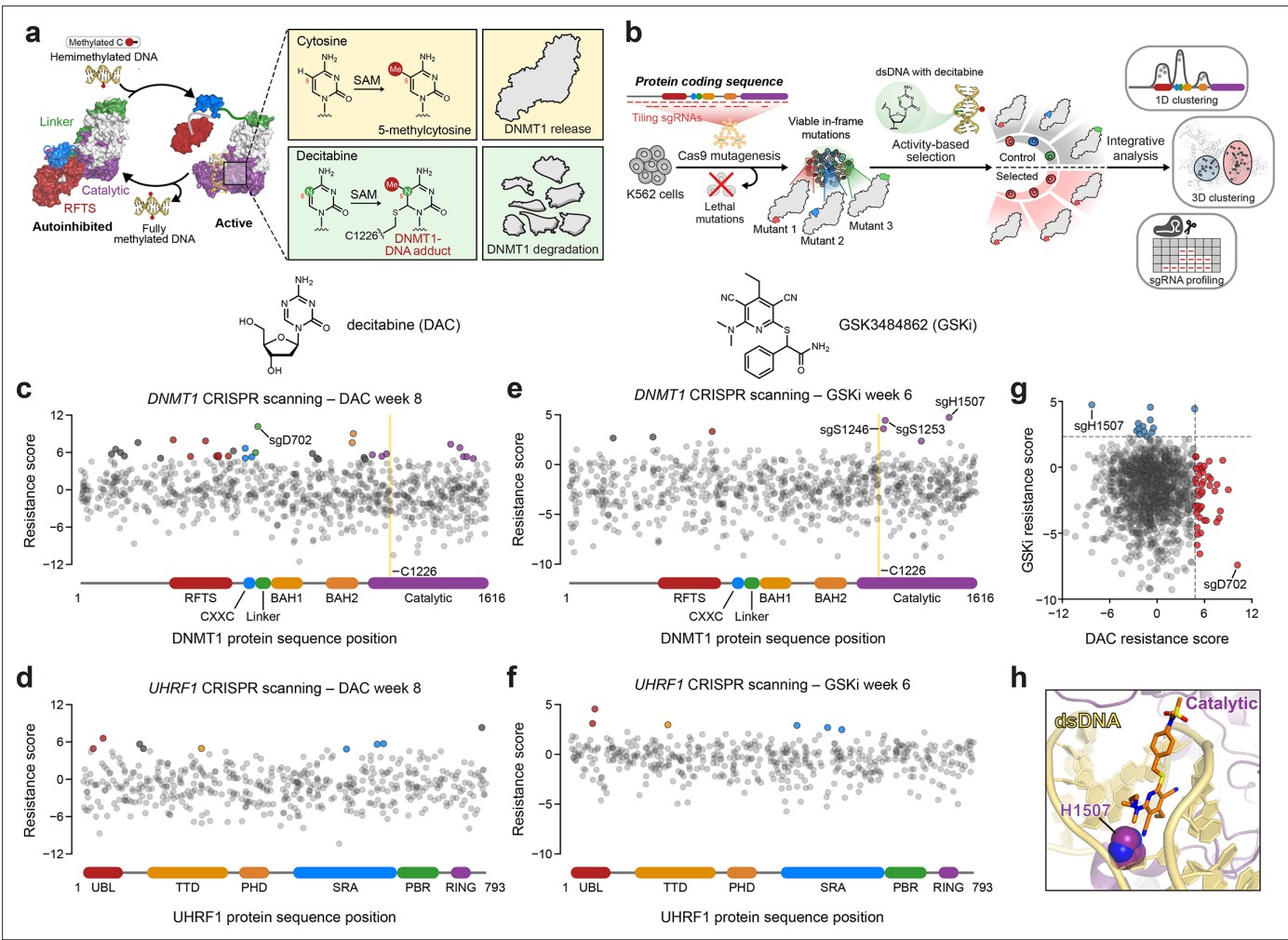

**Figure 1.** Activity-based CRISPR scanning of *DNMT1* and *UHRF1*. (**a**) Schematic showing surface representations of the autoinhibited (PDB: 4WXX) and active (PDB: 4DA4) conformations of DNMT1 and DNMT1-mediated methylation of cytosine and decitabine (DAC). Methylation of DAC leads to the formation of a covalent DNMT1–DNA adduct and subsequent proteasomal degradation. SAM, *S*-adenosyl-L-methionine. (**b**) Schematic of the activity-based CRISPR scanning workflow. K562 cells were transduced with a pooled single-guide RNA (sgRNA) library targeting both *DNMT1* and *UHRF1* and treated with vehicle or DAC for 8 weeks. DAC treatment was performed at 100 nM for 5 weeks, followed by 1 μM for 3 weeks. For CRISPR-suppressor scanning experiments using GSK3484862 (GSKi), cells were treated with vehicle or GSKi for 6 weeks. GSKi treatment was performed at 1 μM for 3 weeks, followed by 5 μM for 3 weeks. Genomic DNA was isolated from vehicle- and drug-treated cells and sequenced to determine sgRNA abundance. Scatter plots showing resistance scores (*y*-axis) for sgRNAs targeting *DNMT1* (**c, e**, n = 830) and *UHRF1* (**d, f**, n = 475) in K562 cells after 8 weeks of DAC treatment (**c, d**) or 6 weeks of GSK3484862 (GSKi) treatment (**e, f**). Resistance scores were calculated as the log2(fold-change) of sgRNA frequencies in drug versus vehicle treatment, followed by normalization to the mean of the negative control sgRNAs (n = 77). The sgRNAs are arrayed by amino acid position in the *DNMT1* and *UHRF1* coding sequences (*x*-axis) corresponding to the positions of their predicted cut sites. Protein domains are demarcated by the colored bars along the *x*-axis. The yellow bands in (**c**) and (**e**) demarcate the position of the catalytic cysteine (C1226) in the DNMT1 active site. Data points represent the mean resistance score across three replicate treatments. Enriched sgRNAs with resistance scores greater than 2 standard deviations (SDs) above the mean of the negative control sgRNAs are colored by their corresponding domain. (**g**) Scatter plot showing resistance scores of *DNMT1*- and *UHRF1*-targeting sgRNAs after 8 weeks of DAC treatment (*x*-axis) or 6 weeks of GSKi treatment (*y*-axis). Dotted lines represent two SDs above the mean of the negative control sgRNAs. sgRNAs that are enriched in DAC or GSKi treatment are colored in red and blue, respectively. (**h**) Structural view of DNMT1 complexed to DNA (yellow) with GSK3830052 (orange), a structural analog of GSK3484862, highlighting the location of H1507 (spheres) in the catalytic domain (purple), which is targeted by the top enriched sgRNA in the GSKi screen (PDB: 6X9J).

The online version of this article includes the following source data and figure supplement(s) for figure 1:

**Figure supplement 1.** Activity-based CRISPR scanning of *DNMT1* and *UHRF1*.

**Figure supplement 1—source data 1.** Raw unedited blot image for the blot shown in *Figure 1—figure supplement 1B*.

**Figure supplement 1—source data 2.** Annotated unedited blot image for the blot shown in *Figure 1—figure supplement 1B*.

intervals displaying greater enrichment than expected by chance. In brief, we used LOESS regression (*Schoonenberg et al., 2018*; *Sher et al., 2019*) to estimate per-residue resistance scores from sgRNA resistance scores and compared them to a simulated distribution generated by shuffling sgRNA resistance scores (*Figure 2a*, see Methods). We identified three linear clusters of enriched residues in *DNMT1*, spanning amino acids (aa) 119–147 in the N-terminus, aa518–571 in the RFTS, and aa652–701 in the CXXC and linker regions (*Figure 2b, c*).

Notably, aa518–571 reside in the C-terminal lobe of the RFTS that interfaces with the catalytic domain and aa652–701 span most of the CXXC and part of the CXXC–BAH1 linker, both of which are implicated in DNMT1 autoinhibition (*Figure 2c*; *Song et al., 2011*; *Takeshita et al., 2011*; *Zhang et al., 2015*). Finally, aa119–147 reside within the disordered N-terminus, which remains structurally unresolved and poorly characterized.

Functional protein regions can comprise spatially proximal residues that are distal on the linear CDS. Spatial clustering of cancer mutations at such regions is often used as evidence of protein function or positive selection (*Kamburov et al., 2015*; *Martínez-Jiménez et al., 2020*). Therefore, we next considered whether sgRNA enrichment patterns might emerge in 3D space that are not readily observed on the linear CDS. To assess 3D sgRNA clustering, we refined and applied a structure-guided clustering methodology that we previously adapted (*Vinyard et al., 2019*; *Kamburov et al., 2015*). In brief, we calculated proximity-weighted enrichment scores (PWES) for all pairwise combinations of resolved sgRNAs using (1) their resistance scores and (2) the Euclidean distance between their targeted residues. Then, we performed hierarchical clustering on the resultant PWES matrix to define clusters of spatially proximal sgRNAs with similar PWES profiles (*Figure 2d*, see Methods).

For 3D sgRNA clustering, we used the structure of autoinhibited human DNMT1$_{351-1600}$ (PDB: 4WXX) (*Zhang et al., 2015*). This structure resolves the greatest number of residues and is the only human DNMT1 structure that includes the RFTS domain, which is involved in DNMT1 autoinhibition and implicated in our 1D clustering analysis (*Figure 2b*). We calculated PWES profiles for the 646 sgRNAs targeting resolved DNMT1 residues and performed hierarchical clustering, identifying 19 clusters of sgRNAs with varying mean resistance scores (*Figure 2e–g*, *Figure 2—figure supplement 1a, b*). The top enriched cluster, cluster 1, comprised many sgRNAs targeting the same intervals in the RFTS, CXXC, and linker regions previously found in our 1D clustering analysis (*Figure 2b, f*). Strikingly, these enriched cluster 1 residues span a multi-domain contact interface that is critical for mediating DNMT1 autoinhibition, suggesting that the prior linear clusters correspond to a singular 3D hotspot (*Figure 2c, f*; *Song et al., 2011*; *Takeshita et al., 2011*; *Zhang et al., 2015*). By contrast, the second-most enriched cluster, cluster 2, mainly comprised sgRNAs targeting a region of the catalytic domain, with two additional sgRNAs targeting the BAH2 domain (*Figure 2c, f*). Taken together, our findings suggest that enriched sgRNAs may target regions of DNMT1 that regulate its activity.

## Cluster 1 mutations in the RFTS, CXXC, and linker regions enhance DNMT1 activity

We next investigated cluster 1 in further detail due to its consistent enrichment across our analyses. In particular, we considered (1) the overall composition and frequency of mutations found in cluster 1 sgRNAs and (2) their functional consequences at the protein level. To assess the underlying mutational outcomes, we individually transduced K562 cells with eight enriched cluster 1 sgRNAs targeting distinct DNMT1 regions distally positioned on the CDS (*Figure 3a*). Transduced cells were treated with vehicle or DAC for 8 weeks and genotyped. Raw sequencing data were processed and aligned with CRISPResso2 (*Clement et al., 2019*) to identify DNA variants and quantify their frequencies. We then characterized variants and their impact at the protein level with a custom pipeline (*Figure 3—figure supplement 1a*, see Methods). Briefly, each DNA variant was classified by the size, location, and sequence context of its mutation, and those classified as in-frame were realigned, trimmed, and translated. For downstream analyses, variants were classified into three major categories: wild-type, in-frame, and loss-of-function. Coding variants retaining the reference protein sequence (e.g., unedited, silent mutations) were considered wild-type. Coding mutations that maintain the reading frame, excluding nonsense and wild-type alleles, were classified as in-frame. Variants predicted to encode a non-functional protein product were classified as loss-of-function (i.e., frameshift, nonsense, splice site mutations).

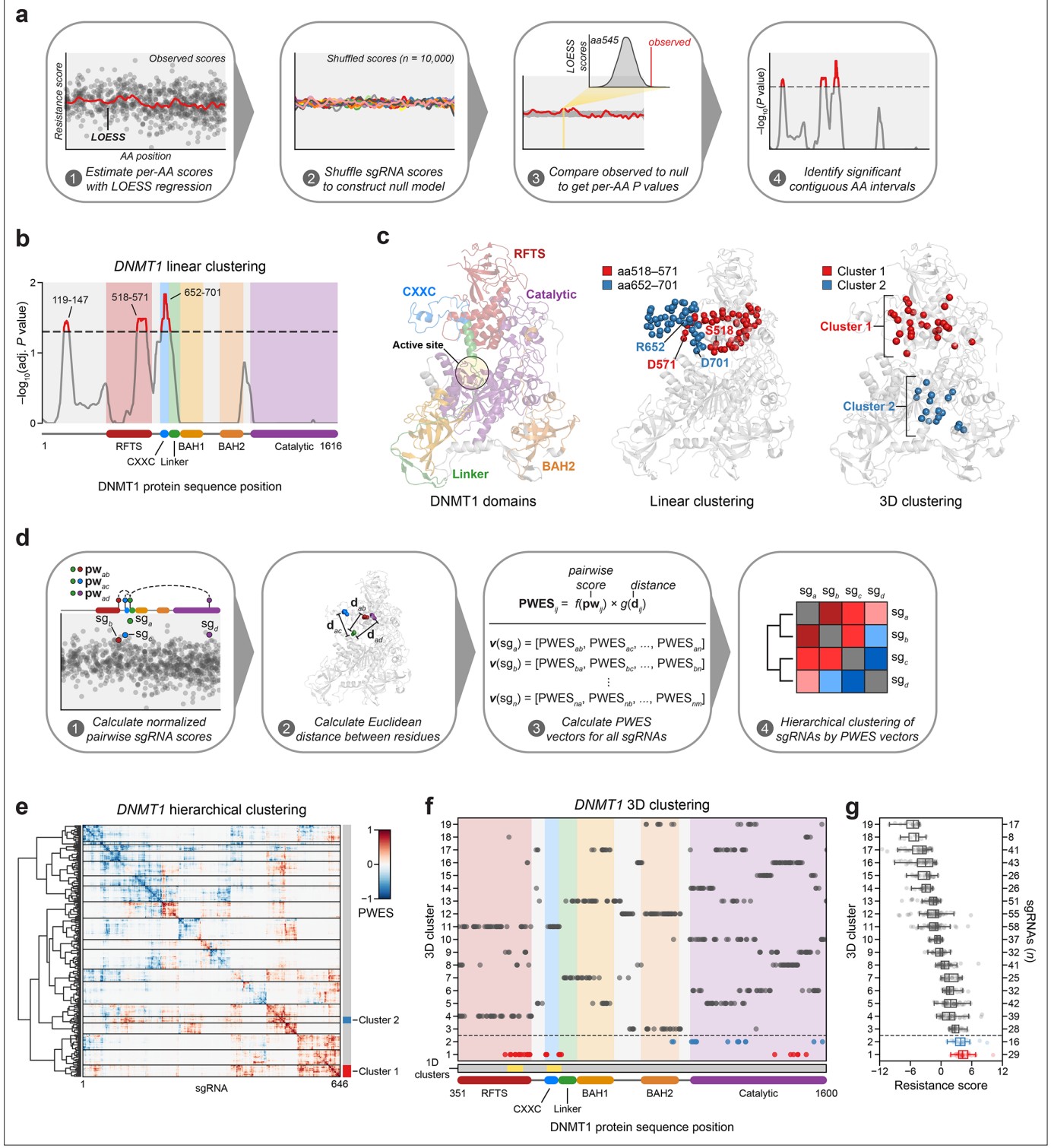

**Figure 2.** Linear and spatial clustering of CRISPR scanning data identifies putative hotspots in DNMT1 that modulate enzyme activity. (**a**) Overview of the linear clustering workflow. LOESS regression and interpolation are applied to the single-guide RNA (sgRNA) resistance scores to estimate per-residue resistance scores for the entire protein coding sequence. sgRNA scores are then shuffled ($n$ = 10,000 iterations) and per-residue resistance scores are recalculated to simulate a null model of random enrichment. Empirical $P$ values for each amino acid are determined by comparing the observed score to the simulated distribution and adjusted with the Benjamini–Hochberg procedure. Linear clusters are defined as contiguous intervals of amino acids with adjusted $P$ values ≤0.05. (**b**) Line plot showing $-\log_{10}$-transformed adjusted $P$ values for the observed per-residue resistance scores ($y$-axis) plotted against the *DNMT1* CDS ($x$-axis). DNMT1 domains are demarcated by the colored background and bars along the $x$-axis. The dotted

*Figure 2 continued on next page*

*Figure 2 continued*

line corresponds to P = 0.05 and residues with P ≤ 0.05 are highlighted in red with linear clusters annotated. (**c**) Structural views of autoinhibited DNMT1 (PDB: 4WXX) highlighting its domains (left panel), the linear clusters (middle panel) spanning aa518–571 (red spheres) and aa652–701 (blue spheres), and the 3D clusters (right panel) 1 (red spheres) and 2 (blue spheres). The linear cluster spanning aa119–147 is not resolved in the structure. The DNMT1 active site is denoted (yellow circle) in the left panel. (**d**) Overview of the 3D clustering workflow. Normalized pairwise resistance scores for sgRNAs are calculated and then scaled relative to the Euclidean distance between their targeted residues in the structural data to generate the final proximity-weighted enrichment score (PWES) for all possible pairwise sgRNA combinations. Each row or column in the resultant PWES matrix thus represents a vector of PWES values for a single sgRNA against all other sgRNAs targeting resolved residues. Hierarchical clustering is applied to the PWES matrix to group sgRNAs by similarities in their overall PWES profiles. (**e**) Heatmap depicting the PWES matrix of all pairwise combinations of sgRNAs (*n* = 646) targeting resolved residues in the structure of autoinhibited DNMT1 (PDB: 4WXX). sgRNAs are grouped by cluster, with black lines demarcating each cluster on the heatmap. Cluster 1 sgRNAs (*n* = 29) and cluster 2 sgRNAs (*n* = 16) are highlighted in red and blue, respectively. (**f**) Scatter plot showing the targeted amino acid positions of the *DNMT1*-targeting sgRNAs used in the 3D clustering analysis. sgRNAs are grouped by 3D cluster (*y*-axis) derived from (**e**) and plotted against the *DNMT1* CDS (*x*-axis). Clusters are numbered by the mean resistance score of their component sgRNAs, with cluster 1 representing the greatest mean resistance score. Clusters 1 and 2 are highlighted in red and blue, respectively. Amino acid intervals corresponding to the linear clusters in (**b**) are highlighted in yellow. (**g**) Box plot showing the sgRNA resistance scores (*x*-axis) for each of the 3D clusters (*y*-axis, left) derived from (**e, f**). The number of sgRNAs per cluster is shown on the *y*-axis (right) and individual sgRNAs within each cluster are shown as points. Clusters 1 and 2 are highlighted in red and blue, respectively. The central band, box boundaries, and whiskers represent the median, interquartile range (IQR), and 1.5 × IQR, respectively.

The online version of this article includes the following figure supplement(s) for figure 2:

**Figure supplement 1.** Linear and spatial clustering of CRISPR scanning data identifies putative hotspots in DNMT1 that modulate enzyme activity.

We observed a dramatic enrichment of in-frame mutations and concomitant depletion of the wild-type allele under DAC treatment for most cluster 1 sgRNAs (*Figure 3b, c*), supporting the notion that in-frame mutations in cluster 1 confer a fitness advantage to cells under DAC treatment. We next considered whether cluster 1 in-frame variants disrupt critical interactions that mediate DNMT1 auto-inhibition. To explore this possibility, we identified the top enriched in-frame variants across cluster 1 sgRNAs, excluding sgT1503 and sgG1504/N1505 due to their lack of enriched in-frame variants (*Figure 3d–f*, *Supplementary file 2*). Overall, enriched in-frame variants were primarily deletions ranging from −2 to −8 aa (*Figure 3—figure supplement 1b, c*). We first examined sgD702 as it exhibited an abundance of in-frame mutations (>70% in DAC and >50% in vehicle) and significant depletion of the wild-type allele (1.2% in DAC versus 26.4% in vehicle). Enriched in-frame mutations in sgD702 likely disrupt an α-helix in the linker, which includes a stretch of acidic residues (D700–E703) that contact the RFTS and catalytic domains to promote autoinhibition (*Song et al., 2011*; *Takeshita et al., 2011*; *Zhang et al., 2015*; *Figure 3f*). In fact, one of the top enriched in-frame variants, M694_D701del, was identical to a previously characterized overactive DNMT1 mutant (*Zhang et al., 2015*).

Prompted by these observations, we sought to determine whether other enriched cluster 1 variants also structurally disrupt DNMT1 autoinhibition. Indeed, many of the top enriched variants perturbed key residues that mediate inter-domain contacts. For example, enriched in-frame mutations in the CXXC domain generated by sgR652 deleted up to 8 aa spanning F648–C656. This region contacts the linker (*Figure 3f*, right panel) and includes a conserved patch of basic residues (K649–R652) that when mutated has been shown to increase DNMT1 activity (*Bashtrykov et al., 2012*). Similarly, observed mutations in the RFTS domain also perturbed residues spanning three discrete intervals across this interface (D526–E532, L542–A554, and M581–L592), despite their distance on the CDS. These intervals encompass α-helices that make extensive polar contacts with the linker and catalytic domains (vide infra). For sgI531, enriched in-frame variants likely compromise polar contacts with residues in the TRD, formed by the side chains of E525, D526, and E532 (*Figure 3e*, left panel).

Likewise, in-frame variants enriched in sgF545, sgT546, and sgI585 also presumably disrupt similar interactions with the TRD (E547, D548, and D583) and the CXXC–BAH1 linker (R582 and K586) (*Figure 3e, f*). Supporting these results, prior biochemical studies have demonstrated that DNMT1 mutations at analogous or proximal positions exhibit increased methyltransferase activity compared to wild-type DNMT1 (*Zhang et al., 2015*; *Bashtrykov et al., 2012*; *Bashtrykov et al., 2014b*; *Berkyurek et al., 2014*; *Dolen et al., 2019*). Taken together, our findings strongly suggest that cluster 1 sgRNAs may confer a selective advantage to DAC by generating gain-of-function mutations that relieve DNMT1 autoinhibition and enhance its enzymatic activity.

To confirm whether these in-frame mutations indeed enhance DNMT1 activity, we biochemically characterized a subset of enriched cluster 1 mutations in the RFTS, CXXC, and linker regions,

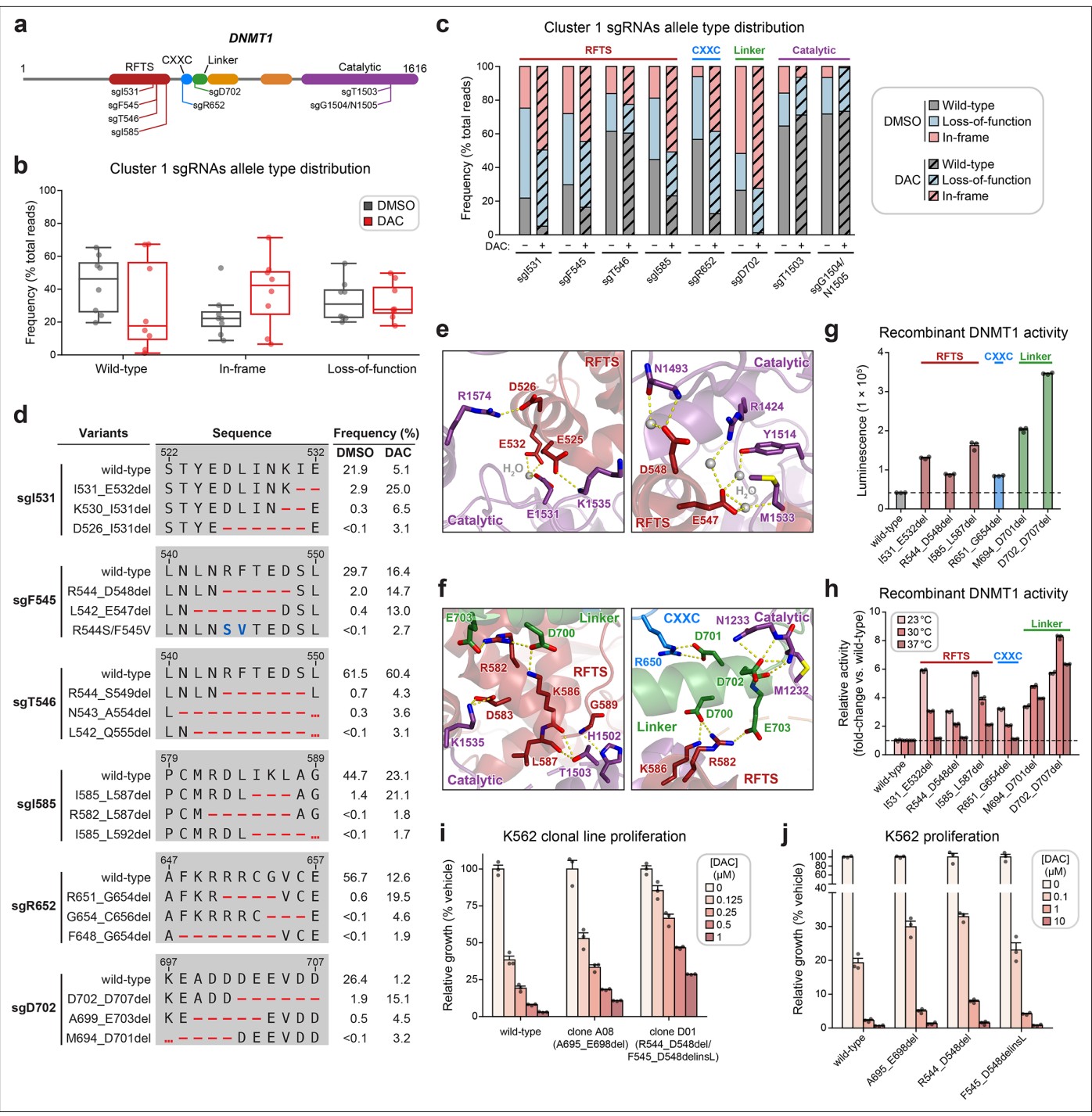

**Figure 3.** Cluster 1 single-guide RNAs (sgRNAs) generate hypermorphic *DNMT1* mutations in the RFTS, CXXC, and linker regions that abrogate autoinhibition. (**a**) Schematic showing the amino acid positions on the *DNMT1* CDS targeted by selected cluster 1 sgRNAs (*n* = 8). (**b**) Box plots showing the observed frequencies (percentage of total reads, *y*-axis) of wild-type, in-frame, and loss-of-function alleles (*x*-axis) after 8 weeks of treatment with vehicle (gray) or 100 nM decitabine (DAC, red) for the cluster 1 sgRNAs shown in (**a**). Individual sgRNAs are shown as points. The central band, box boundaries, and whiskers represent the median, interquartile range (IQR), and 1.5 × IQR, respectively. (**c**) Stacked bar plot showing the observed frequencies (percentage of total reads, *y*-axis) of wild-type, in-frame, and loss-of-function alleles for selected cluster 1 sgRNAs (*x*-axis) after 8 weeks of treatment with vehicle or 100 nM DAC. (**d**) Table showing the amino acid sequence alignment and observed frequencies (percentage of total reads) of the wild-type and top enriched in-frame variants across cluster 1 sgRNAs after 8 weeks of treatment with vehicle or 100 nM DAC. In-frame variants were considered enriched if the observed frequency in DAC treatment was ≥1% and the log$_2$(fold-change) of the observed frequency in DAC versus vehicle treatment was ≥2. Enriched in-frame variants meeting these criteria were sorted by their observed frequency in DAC treatment and the top

*Figure 3 continued on next page*

*Figure 3 continued*

3 most abundant are shown. sgT1503 and sgG1504/N1505 were excluded due to the lack of enriched in-frame variants. Amino acid deletions are represented as red dashes and substitutions are highlighted in blue. Red ellipses are used to denote amino acid deletions that exceed the length of the shown sequence alignment. (**e, f**) Structural views of various regions of the DNMT1 autoinhibitory interface (PDB: 4WXX) perturbed by enriched in-frame variants (from **d**) generated by cluster 1 sgRNAs. The panels highlight key inter-domain interactions disrupted by sgI531 (**e**, left), sgF545/sgT546 (**e**, right), and sgI585/sgR652/sgD702 (**f**). Key residues in the RFTS (red), CXXC (blue), linker (green), and catalytic (purple) domains that mediate polar contacts are shown as sticks. Hydrogen bonds are represented by dotted yellow lines and water molecules are depicted as gray spheres. (**g**) Bar plot showing recombinant DNMT1 enzyme activity (luminescence, *y*-axis) for wild-type DNMT1 and selected cluster 1 variants (from **d**, *x*-axis) in the luminescence-based MTase-Glo assay. Wild-type is depicted in gray and variants are colored according to the domain in which the mutation is located. The dotted line represents the mean luminescence of the wild-type DNMT1 construct. (**h**) Bar plot showing the relative enzyme activity (*y*-axis) of recombinant DNMT1 constructs (*x*-axis) in the MTase-Glo assay at the indicated temperatures. Relative enzyme activity was calculated as the fold-change in luminescence for the given construct relative to the wild-type construct at the specified temperature. The dotted line represents the mean activity of the wild-type construct. (**i**) Bar plot showing the relative cellular proliferation (*y*-axis) of wild-type and clonal *DNMT1*-mutated K562 cells (*x*-axis) treated with vehicle or DAC for 7 days. Relative cellular proliferation was calculated as the percent growth relative to vehicle treatment. (**j**) Bar plot showing the relative cellular proliferation (*y*-axis) of K562 cells treated with vehicle or DAC for 7 days following lentiviral transduction with vectors expressing a short hairpin RNA targeting endogenous *DNMT1* transcripts and the specified DNMT1 construct (*x*-axis). Relative cellular proliferation was calculated as the percent growth relative to vehicle treatment. For bar plots in (**g–j**), bars represent the mean ± standard error of the mean (SEM) across three replicates and one of two independent experiments is shown.

The online version of this article includes the following source data and figure supplement(s) for figure 3:

**Figure supplement 1.** Cluster 1 single-guide RNAs (sgRNAs) generate hypermorphic *DNMT1* mutations in the RFTS, CXXC, and linker regions that abrogate autoinhibition.

**Figure supplement 1—source data 1.** Raw unedited blot image for the blot shown in *Figure 3—figure supplement 1H*.

**Figure supplement 1—source data 2.** Annotated unedited blot image for the blot shown in *Figure 3—figure supplement 1H*.

**Figure supplement 1—source data 3.** Raw unedited blot image for the blot shown in *Figure 3—figure supplement 1I*.

**Figure supplement 1—source data 4.** Annotated unedited blot image for the blot shown in *Figure 3—figure supplement 1I*.

including the previously validated overactive M694_D701del mutation (*Figure 3d*). We purified recombinant wild-type and mutant DNMT1$_{351-1616}$ constructs and evaluated their enzymatic activity (*Dolen et al., 2019*; *Hsiao et al., 2016*). Corroborating our structural predictions, these mutants exhibited 1.7- to 5.8-fold increased activity compared to wild-type DNMT1$_{351-1616}$ (*Figure 3g*). To gain further insight into the increased activity of cluster 1 mutants, we evaluated their methyltransferase activity at multiple temperatures (*Figure 3h*). Whereas the autoinhibitory linker mutants were consistently more active than wild-type DNMT1, the RFTS and CXXC mutants exhibited a temperature-dependent decrease in relative activity compared to wild-type DNMT1. RFTS and CXXC mutants were three- to sixfold more active than wild-type at 23°C but exhibited comparable activity to wild-type at 37°C, apart from I585_L587del. Our results are in accordance with previous biochemical studies reporting similar temperature-dependent decreases in the relative activity of RFTS mutants compared to wild-type DNMT1 (*Berkyurek et al., 2014*), supporting a model where disruption of key contacts at the DNMT1 autoinhibitory interface lowers the activation energy barrier required to release the RFTS from the catalytic center. However, we cannot rule out the possibility of alternative mechanisms for the observed temperature-dependent trend in relative activity, such as reduced stability of the RFTS mutants (vide infra). Interestingly, the observed hyperactivity of the autoinhibitory linker mutants across increasing temperatures suggests these mutations may disrupt DNMT1 autoinhibition through distinct biochemical mechanisms.

As reducing DNMT1 protein levels attenuates DNMT1–DNA adduct-induced cytotoxicity, we next considered whether cluster 1 mutations influence DNMT1 protein abundance and/or DAC-induced depletion of DNMT1 (*Patel et al., 2010*). To address these questions, we evaluated cellular DNMT1 levels with a fluorescent reporter system (*Sievers et al., 2018*; *Słabicki et al., 2020*) in which full-length DNMT1 is fused to an EGFP–IRES–mCherry cassette. EGFP acts as a proxy for DNMT1 protein levels while mCherry serves as an internal standard, accounting for cell-to-cell variation in reporter integrations, expression, or homeostasis. We transduced K562 cells with reporter DNMT1 constructs encoding wild-type and mutant DNMT1 and assessed EGFP/mCherry fluorescence ratios after 3 days of treatment with vehicle or DAC (*Figure 3—figure supplement 1d–f*). Under vehicle treatment, EGFP/mCherry ratios for the CXXC and linker mutants were comparable to wild-type DNMT1, whereas RFTS mutants displayed significantly lower EGFP/mCherry ratios, suggesting that RFTS mutations

may destabilize the protein (*Figure 3—figure supplement 1d*). While we note, however, that the use of EGFP fusion constructs may confound precise measurements of DNMT1 stability due to the long half-life of EGFP, our results are consistent with previous studies of clinical *DNMT1* hotspot mutations found in hereditary sensory and autonomic neuropathy type 1E (HSAN1E) and autosomal dominant cerebellar ataxia, deafness, and narcolepsy (ADCA-DN) patients, demonstrating that mutations in the RFTS domain destabilize DNMT1 (*Klein et al., 2011*; *Winkelmann et al., 2012*; *Smets et al., 2017*). Upon treatment with DAC, cluster 1 mutants were degraded at similar levels as wild-type DNMT1, suggesting that these mutations are unlikely to confer a selective advantage through resistance to degradation (*Figure 3—figure supplement 1e*).

As an orthogonal means of validation and characterization, we isolated two clonal cell lines from the activity-based CRISPR scanning experiment containing endogenous *DNMT1* mutations (*Figure 3—figure supplement 1g*). Clone A08 contains an autoinhibitory linker mutation (A695_E698del) and clone D01 harbors heterozygous RFTS mutations (R544_D548del; F545_D548delinsL), one of which we note is identical to one of our selected cluster 1 mutations (*Figure 3d*, *Figure 3—figure supplement 1g*). We observed a partial growth rescue in both clones compared to wild-type K562 in the presence of DAC (*Figure 3i*), supporting the notion that mutations perturbing DNMT1 autoinhibition confer resistance. In agreement with our fluorescent reporter results, we observed DAC-mediated degradation of DNMT1 in both clones, suggesting that these mutations are unlikely operate through resistance to degradation (*Figure 3—figure supplement 1h*).

To ensure that the observed resistance in the clonal lines was not driven by clone-specific properties independent of their *DNMT1* mutations, we generated K562 cell lines expressing wild-type DNMT1 or the individual clone-derived mutants. Simultaneously, we selectively knocked down endogenous *DNMT1* by transducing a short hairpin RNA (shRNA) targeting the 3' untranslated region (UTR) of *DNMT1* (*Figure 3—figure supplement 1i*). Consistent with our data from the clonal lines, cells expressing the autoinhibitory linker and RFTS mutants all exhibited partial resistance to DAC compared those expressing the wild-type DNMT1 construct (*Figure 3j*). Altogether, our results collectively support a model in which cluster 1 mutations confer partial resistance to DAC and operate primarily by disrupting DNMT1 autoinhibition to enhance enzymatic activity.

## Integrative analysis reveals distinct mutational profiles between cluster 1 and 2 sgRNAs

As our approach accurately identified a validated mechanism of DNMT1 autoinhibition, we next investigated how cluster 2 sgRNAs operate. We evaluated mutational outcomes generated by 10 of the top enriched cluster 2 sgRNAs targeting the BAH2 and catalytic domains (*Figure 4a*, *Supplementary file 2*). Like before, K562 cells were individually transduced, treated with vehicle or DAC for 8 weeks, and genotyped. Notably, observed mutational outcomes were highly dissimilar between cluster 1 and 2 sgRNAs. Overall, cluster 2 sgRNAs exhibited preferential enrichment of the wild-type allele and loss-of-function variants with concomitant depletion of in-frame variants, in stark contrast to the substantial enrichment of in-frame mutations seen in cluster 1 sgRNAs (*Figure 4b*). Notably, the enrichment patterns of cluster 1 sgRNAs targeting the catalytic domain (sgT1503 and sgG1504/N1505) resembled those of cluster 2 sgRNAs (*Figure 3c*). Based on these observations, we considered whether the prominent differences in their mutational profiles might (1) enable us to classify individual sgRNAs by the features of their mutational profiles and (2) indicate that clusters 1 and 2 operate through distinct mechanisms.

Subsequently, we first explored whether we could distinguish individual sgRNAs through characteristics of their mutational profiles. In addition to sgRNA resistance scores, we included metrics such as the absolute frequencies (percentage of total reads) of wild-type and in-frame alleles and the relative frequency (percentage of edited reads) of in-frame alleles. Because mutational outcomes across sgRNAs can be highly complex, we also included additional features to represent the similarity and diversity of mutational outcomes. To assess mutational profile similarity in vehicle versus DAC treatment, we calculated the Pearson correlation of observed allele frequencies and the symmetric Kullback–Leibler (KL) divergence, which quantifies the similarity of two probability distributions with greater values indicating greater dissimilarity. To measure mutational diversity under DAC treatment, we calculated the Gini coefficient, a statistical measure of dispersion, with respect to all alleles and edited alleles. Finally, to capture the directionality of change across treatments, we calculated the

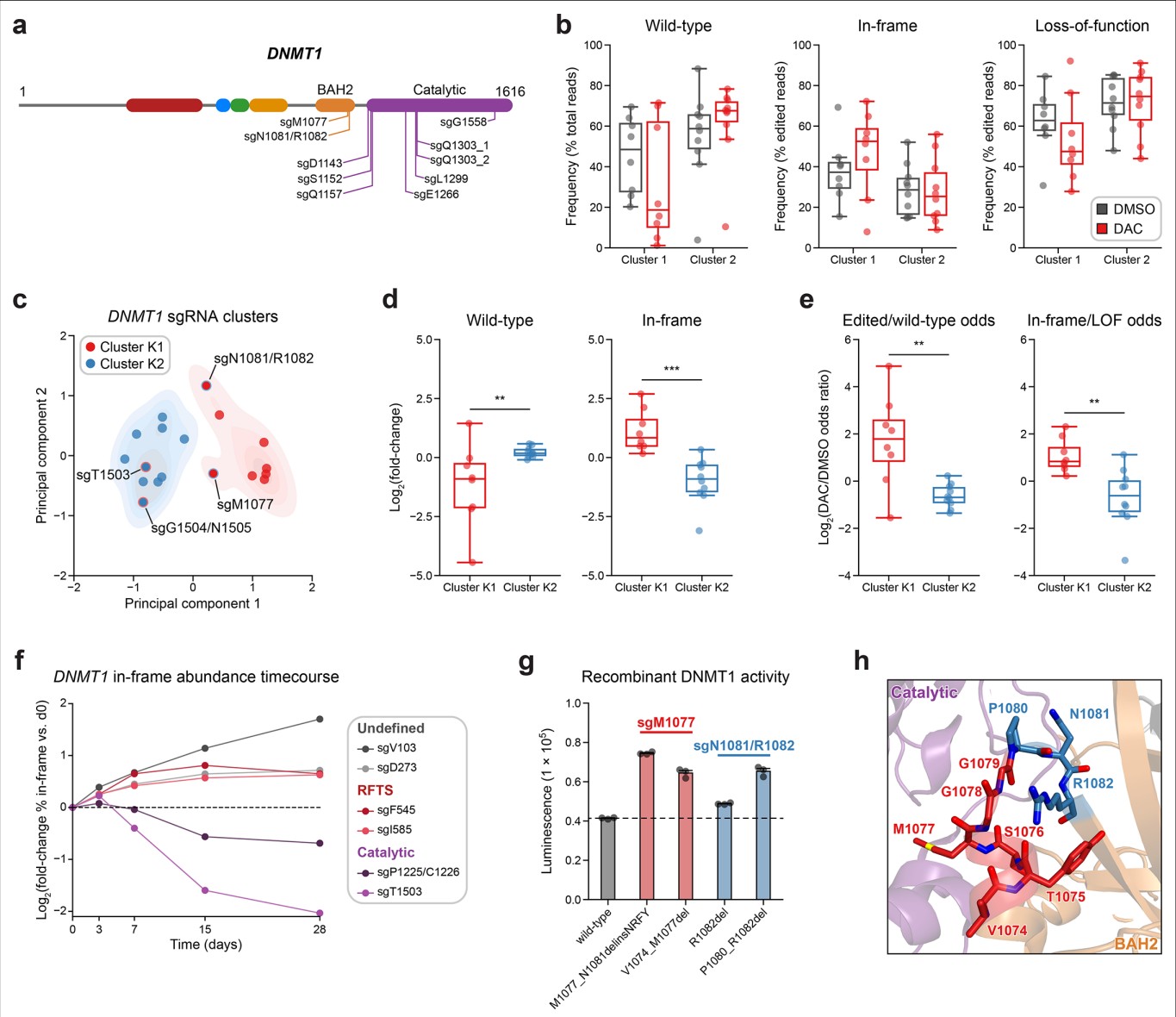

**Figure 4.** Integrative analysis reveals distinct mutational profiles between cluster 1 and 2 single-guide RNAs (sgRNAs). (**a**) Schematic showing the amino acid positions on the *DNMT1* CDS targeted by selected cluster 2 sgRNAs (*n* = 10). (**b**) Box plots showing the observed frequencies (*y*-axis) of wild-type, in-frame, and loss-of-function (LOF) alleles (*x*-axis) after 8 weeks of treatment with vehicle (gray) or 100 nM decitabine (DAC, red) for cluster 1 (*n* = 8, from *Figure 3a*) and cluster 2 sgRNAs (*n* = 10, from **a**). Wild-type allele frequencies are plotted as the percentage of total reads. In-frame and loss-of-function allele frequencies are plotted as the percentage of edited (i.e., non-wild-type) reads. (**c**) Scatter plot showing *DNMT1*-targeting sgRNAs from clusters 1 and 2 (*n* = 18) projected onto principal component space after principal component analysis on features of their mutational profiles. The sgRNAs were partitioned using *k*-means clustering (*k* = 2) to identify clusters K1 (red) and K2 (blue), corresponding primarily to the original 3D clusters 1 and 2. sgRNAs reassigned from cluster 1 to K2 or cluster 2 to K1 are annotated and denoted with red and blue borders, respectively. Contours depict a bivariate kernel density estimation for clusters K1 (red) and K2 (blue). (**d**) Box plots showing the log$_2$(fold-change) enrichment (*y*-axis) of wild-type (left) and in-frame variants (right) in DAC versus vehicle treatment for cluster K1 (*n* = 8, red) and K2 (*n* = 10, blue) sgRNAs. Log$_2$(fold-change) was calculated based on the observed frequency (percentage of total reads) in DAC versus vehicle treatment. (**e**) Box plots showing the log$_2$(odds ratio) in DAC versus vehicle treatment (*y*-axis) for edited versus wild-type odds (left) and in-frame versus LOF odds (right) for cluster K1 (red) and K2 (blue) sgRNAs. Edited/wild-type odds were calculated using absolute frequencies (percentage of total reads) and in-frame/LOF odds were calculated using relative frequencies (percentage of edited reads). (**f**) Line plot showing the relative abundance of in-frame mutations (*y*-axis) over time (*x*-axis) for individual sgRNAs (*n* = 6) targeting the DNMT1 N-terminus ('Undefined', gray), RFTS (red), and catalytic domains (purple). Relative in-frame abundance was calculated as the log$_2$(fold-change) of the absolute frequency (percentage of total reads) of in-frame variants at the indicated time point versus day 0 and is represented by the colored points. Mutations were considered in-frame if they preserved the coding frame and did not disrupt a splice site or result in a premature stop codon. The dotted line corresponds to the relative abundance of in-frame mutations at day 0. (**g**) Bar plot showing recombinant DNMT1 enzyme

*Figure 4 continued on next page*

*Figure 4 continued*

activity (luminescence, *y*-axis) for wild-type DNMT1 and selected BAH2 variants (*x*-axis) enriched in sgM1077 (red) and sgN1081/R1082 (blue) in the luminescence-based MTase-Glo assay. The dotted line represents the mean luminescence of the wild-type DNMT1 construct. Bars represent the mean ± standard error of the mean (SEM) across three replicates. One of two independent experiments is shown. (**h**) Structural view of the DNMT1 BAH2 (yellow) region highlighting residues targeted by sgM1077 (red) and sgN1081/1082 (blue). The catalytic domain is shown in purple. Perturbed residues are shown as sticks and annotated (PDB: 4WXX). For box plots in (**b, d, e**) the individual sgRNAs are plotted as points and the central band, box boundaries, and whiskers represent the median, interquartile range (IQR), and 1.5 × IQR, respectively. *P* values (**$P \leq 0.01$; ***$P \leq 0.001$) were calculated with two-sided Mann–Whitney tests.

The online version of this article includes the following figure supplement(s) for figure 4:

**Figure supplement 1.** Integrative analysis reveals distinct mutational profiles between cluster 1 and 2 single-guide RNAs (sgRNAs).

$\log_2$(fold-change) of wild-type and in-frame allele frequencies, as well as the 'odds ratio' of edited/wild-type odds and in-frame/loss-of-function odds in DAC versus vehicle treatment. We defined the edited/wild-type odds as the proportion (percentage of total reads) of edited versus wild-type alleles and in-frame/loss-of-function odds as the proportion (percentage of edited reads) of in-frame versus loss-of-function alleles.

Using these features, we performed principal component analysis (PCA) and *k*-means clustering (*k* = 2) on the *DNMT1*-targeting sgRNAs to partition them by their mutational profiles and evaluate their similarity to our structure-derived clusters. Overall, the resultant *k*-means clusters, which we term clusters K1 and K2 to denote their resemblance to the 3D structure-derived clusters 1 and 2, respectively, largely preserved the sgRNA compositions of their corresponding structure-derived clusters (*Figure 4c*). Notably, the TRD-targeting sgRNAs (sgT1503 and sgG1504/N1505) from cluster 1 were reassigned to cluster K2, in agreement with our previous observations. Conversely, our analysis also reassigned the BAH2-targeting sgRNAs from cluster 2 (sgM1077 and sgN1081/R1082) to cluster K1, suggesting that they share greater resemblance to core cluster 1 sgRNAs and enrich for in-frame gain-of-function variants.

We next evaluated clusters K1 and K2 across the PCA feature set to determine their distinguishing characteristics. Supporting our previous findings, cluster K1 sgRNAs displayed a greater and lower abundance of in-frame and wild-type alleles, respectively, under DAC treatment compared to cluster K2 sgRNAs (*Figure 4—figure supplement 1a*). Similarly, when comparing DAC versus vehicle treatment, cluster K1 sgRNAs exhibited preferential enrichment and depletion of in-frame and wild-type alleles, respectively, and greater edited/wild-type and in-frame/loss-of-function odds ratios relative to cluster K2 sgRNAs (*Figure 4d, e*). Finally, cluster K1 mutational profiles tended to have lower Gini coefficients (all alleles) and Pearson correlations compared to cluster K2 sgRNAs, consistent with the idea that DAC-mediated positive selection of in-frame variants drives greater allelic diversity and mutational divergence relative to no selection in cluster K1 sgRNAs (*Figure 4—figure supplement 1b*). Taken together, our findings support the notion that cluster 1/K1 sgRNAs generate in-frame gain-of-function variants that confer a fitness advantage to DAC.

Due to the (1) preponderance of wild-type and loss-of-function variants observed in cluster 2/K2 sgRNAs (*Figure 4b*, *Figure 4—figure supplement 1a*) and (2) the exclusive catalytic-targeting sgRNA composition of cluster K2 after *k*-means reassignment (*Figure 4c*), we next considered whether these differences may signify an alternative resistance mechanism for cluster 2/K2 sgRNAs distinct from that of cluster 1/K1 sgRNAs. Whereas our data suggest that cluster 1 sgRNAs confer resistance through gain-of-function mutations that enhance DNMT1's catalytic activity, we hypothesized that the relative abundance of loss-of-function mutations and lack of in-frame enrichment found in cluster 2/K2 sgRNAs may be indicative of a knockout-mediated mechanism in which reducing *DNMT1* gene dosage confers resistance to DAC by attenuating DNMT1–DNA adduct-related cytotoxicity.

An obvious question arising from this proposed gene dosage reduction mechanism is why cluster 2/K2 sgRNAs would be particularly predisposed to such an effect, as presumably any *DNMT1*-targeting sgRNA can generate loss-of-function (i.e., frameshift, nonsense, splice site disrupting) mutations. The key insight underlying the proposed mechanism is that sgRNAs targeting essential protein regions and functional domains generate greater proportions of null (i.e., functional knockout) mutations compared to sgRNAs targeting non-essential regions (*Shi et al., 2015*). This occurs because in-frame coding mutations in functionally important protein regions (e.g., DNMT1 catalytic domain) are more likely to disrupt protein function than those found in non-essential regions. Subsequently,

in-frame mutations generated by sgRNAs targeting essential regions are more likely to encode a non-functional protein product and are thus 'effectively loss-of-function'. Importantly, the observation that the specific protein region targeted by an sgRNA influences its likelihood of generating null mutations also implies that (1) not all CDS-targeting sgRNAs are equally effective at driving knockout-mediated effects and (2) sgRNAs that are more effective at generating null mutations may preferentially cluster within functionally important protein regions.

Consequently, we reasoned that cluster 2/K2 sgRNAs, which target the essential catalytic domain, may be more successful at reducing *DNMT1* gene dosage because in-frame mutations in the catalytic domain are more likely to disrupt DNMT1's essential function. That is, cluster 2/K2 sgRNAs may generate greater proportions of such 'effectively loss-of-function' in-frame mutations, and their observed spatial clustering likely reflects the functional importance and mutational intolerance of the catalytic domain. To test this hypothesis, we transduced wild-type K562 cells with 6 individual sgRNAs targeting the DNMT1 N-terminus, RFTS, and catalytic domains and monitored the mutational distribution of the cellular pools over 28 days through targeted amplicon sequencing (*Figure 4f*). We observed increasing frequencies of in-frame mutations over time for sgRNAs targeting the N-terminus and RFTS domain, consistent with the idea that in-frame mutations in these regions are functional and not under strong negative selection. By contrast, catalytic-targeting sgRNAs exhibited considerable depletion of in-frame mutations over time, supporting the notion that in-frame mutations in essential protein regions are functional knockouts and are thus subject to negative selection. These results are in accordance with a mechanism in which catalytic-targeting sgRNAs (e.g., cluster 2/K2) are more effective at conferring DAC resistance through a gene dosage reduction effect due to their enhanced ability to generate null mutations.

Because local DNA sequence context is a primary determinant of an sgRNA's overall distribution of editing outcomes (*Shen et al., 2018*; *Allen et al., 2018*), we next considered whether the observed mutational frequencies in cluster K1 and K2 sgRNAs were consistent with their proposed mechanisms or a product of sequence context biases in Cas9 repair outcomes. We posited that if cluster K1 sgRNAs confer resistance through gain-of-function in-frame mutations, then positive selection for these in-frame variants should result in higher observed frequencies under DAC treatment than their predicted frequencies as repair outcomes. To test this hypothesis, we used inDelphi (*Shen et al., 2018*) to predict editing outcome frequencies from DNA sequence context. We then calculated the predicted in-frame/loss-of-function odds for each sgRNA, as well as the in-frame/loss-of-function odds ratio in DAC versus inDelphi to account for baseline variations in in-frame/loss-of-function odds across sgRNAs. As expected, cluster K1 sgRNAs exhibited significantly higher in-frame/loss-of-function odds under DAC treatment than predicted by inDelphi (*Figure 4—figure supplement 1c*) and significantly higher DAC/inDelphi ratios of in-frame/loss-of-function odds relative to cluster K2 sgRNAs (*Figure 4—figure supplement 1d*), in agreement with the idea that in-frame mutations observed in cluster K1 sgRNAs are under positive selection.

Conversely, our gene dosage reduction hypothesis for cluster K2 sgRNAs relies on their ability to generate 'effectively loss-of-function' in-frame mutations that are functionally equivalent to formal loss-of-function mutations (i.e., frameshift, nonsense, splice site disrupting). This implies that cluster K2 sgRNAs should not exhibit preferential enrichment of in-frame or loss-of-function mutations under DAC treatment and the observed frequencies of these mutations should primarily reflect their predicted probabilities as editing outcomes. Supporting this notion, in-frame/loss-of-function odds in DAC for cluster K2 sgRNAs do not deviate significantly from their expected proportions as predicted by inDelphi (*Figure 4—figure supplement 1c*). Collectively, our results are consistent with a model where (1) in-frame variants found in cluster K1 sgRNAs are positively selected in the presence of DAC and (2) in-frame/loss-of-function variant frequencies in cluster K2 sgRNAs resemble their probabilities as editing outcomes due to being functionally equivalent.

Because our analysis reassigned the cluster 2 BAH2-targeting sgRNAs to cluster K1, we considered whether these sgRNAs also select for in-frame gain-of-function variants under DAC treatment. Consequently, we evaluated the activity of the top 2 enriched in-frame variants generated by sgM1077 (M1077_N1081delinsNRFY, V1074_M1077del) and sgN1081/R1082 (R1082del, P1080_R1082del) (*Figure 4—figure supplement 1e*). Although more modest than cluster 1 mutants, we observed 1.4- to 1.5-fold greater methyltransferase activities across the BAH2 mutants relative to wild-type, apart from R1082del, whose activity was comparable to wild-type (*Figure 4g*). Like the RFTS and CXXC

mutations in cluster 1, we also observed temperature-dependent decreases in relative activity for the BAH2 mutants versus wild-type DNMT1, except for R1082del whose activity was comparable to wild-type DNMT1 across temperatures (*Figure 4—figure supplement 1f*). The BAH2 mutants also displayed similar levels of protein stability and DAC-induced degradation as wild-type DNMT1 in our cellular protein reporter system (*Figure 4—figure supplement 1g, h*). These variants are located near the BAH2–TRD loop (*Figure 4h*), which is thought to restrain the TRD from contacting the DNA substrate in the autoinhibited conformation (*Song et al., 2011*) and we speculate that these BAH2 mutations may perturb mechanisms regulating substrate binding and catalysis. Altogether, our results demonstrate how genotype-level analysis can resolve mutational heterogeneity across sgRNAs within screen-derived clusters, enabling the discovery of new functional protein sites.

## Mutational profiles of individual sgRNAs nominate putative functional regions in UHRF1

UHRF1 is a multifunctional protein that directs DNMT1 to hemimethylated sites during DNA replication. Like DNMT1, UHRF1 is also indispensable for DNA methylation maintenance and ablation of UHRF1 causes global DNA hypomethylation (*Sharif et al., 2007*; *Bostick et al., 2007*). UHRF1-mediated recruitment of DNMT1 to chromatin requires the coordinated function of its various domains (*Xie and Qian, 2018*; *Bronner et al., 2019*). Furthermore, direct and indirect interactions between UHRF1 and DNMT1 not only recruit DNMT1 to chromatin, but also stimulate its activity (*Berkyurek et al., 2014*; *Xie and Qian, 2018*; *Bronner et al., 2019*; *Bashtrykov et al., 2014a*; *Li et al., 2018*; *Harrison et al., 2016*; *Mishima et al., 2020*; *Rothbart et al., 2012*). Consequently, we sought to determine whether our approach could nominate gain-of-function mutations beyond the direct drug target (i.e., DNMT1). As our results indicate that DAC treatment enriches for hypermorphic *DNMT1* mutations, we speculated that DAC treatment may also enrich for mutations in *UHRF1* that influence DNMT1 function.

Although *UHRF1*-targeting sgRNAs were enriched in our activity-based CRISPR scanning screen (*Figure 1d*), the lack of UHRF1 structural data precluded structure-guided clustering analysis. However, our genotype-level analysis of individual sgRNAs suggests that distinct mutational characteristics can indicate functional consequences at the protein level. We therefore reasoned that this approach might identify *UHRF1*-targeting sgRNAs harboring putative gain-of-function mutations based on their mutational profile signatures.

Accordingly, we individually profiled 22 enriched *UHRF1*-targeting sgRNAs in K562 cells under vehicle or DAC treatment for 8 weeks like previously (*Figure 5a*, *Supplementary file 3*). We then performed PCA and *k*-means clustering analysis on the combined dataset of *UHRF1*- and *DNMT1*-targeting sgRNAs with same set of features and number of clusters as before to enable comparisons with DNMT1 clusters K1 and K2 for reference. In particular, we considered (1) how the inclusion of *UHRF1*-targeting sgRNAs might affect the partitioning of *DNMT1*-targeting sgRNAs and (2) whether *UHRF1*-targeting sgRNAs clustering with DNMT1 cluster K1 sgRNAs enrich for gain-of-function variants upon DAC treatment. Our analysis identified two comparably sized clusters (19 and 21 sgRNAs) with 11 *UHRF1*-targeting sgRNAs each (*Figure 5b*). Reassuringly, the inclusion of *UHRF1*-targeting sgRNAs did not alter the clustering of *DNMT1*-targeting sgRNAs, indicating that these clusters may be partitioned analogously to DNMT1 clusters K1 and K2.

To nominate *UHRF1*-targeting sgRNAs that induce potential gain-of-function mutations, we examined the cluster containing DNMT1 cluster K1 sgRNAs, which we term 'drug-divergent'. As the *UHRF1*-targeting sgRNAs doubled the total number of characterized sgRNAs, we first investigated whether drug-divergent sgRNAs were analogous to DNMT1 cluster K1 sgRNAs. Indeed, the mutational profiles of drug-divergent sgRNAs shared similar characteristics to DNMT1 cluster K1 across multiple metrics (*Figure 5—figure supplement 1a–g*) suggesting that they may also enrich for gain-of-function variants under DAC treatment.

We next investigated drug-divergent sgRNAs targeting *UHRF1* to identify in-frame variants enriched in DAC treatment. We first examined sgQ519 and sgK592, which target the SRA domain, as they exhibited the greatest enrichment of in-frame variants and sgRNA resistance score, respectively (*Figure 1d*). The SRA domain specifically binds hemimethylated DNA and is reported to directly interact with and stimulate DNMT1 activity (*Bashtrykov et al., 2014a*; *Arita et al., 2008*; *Avvakumov et al., 2008*; *Hashimoto et al., 2008*). We observed substantial depletion of the wild-type allele and

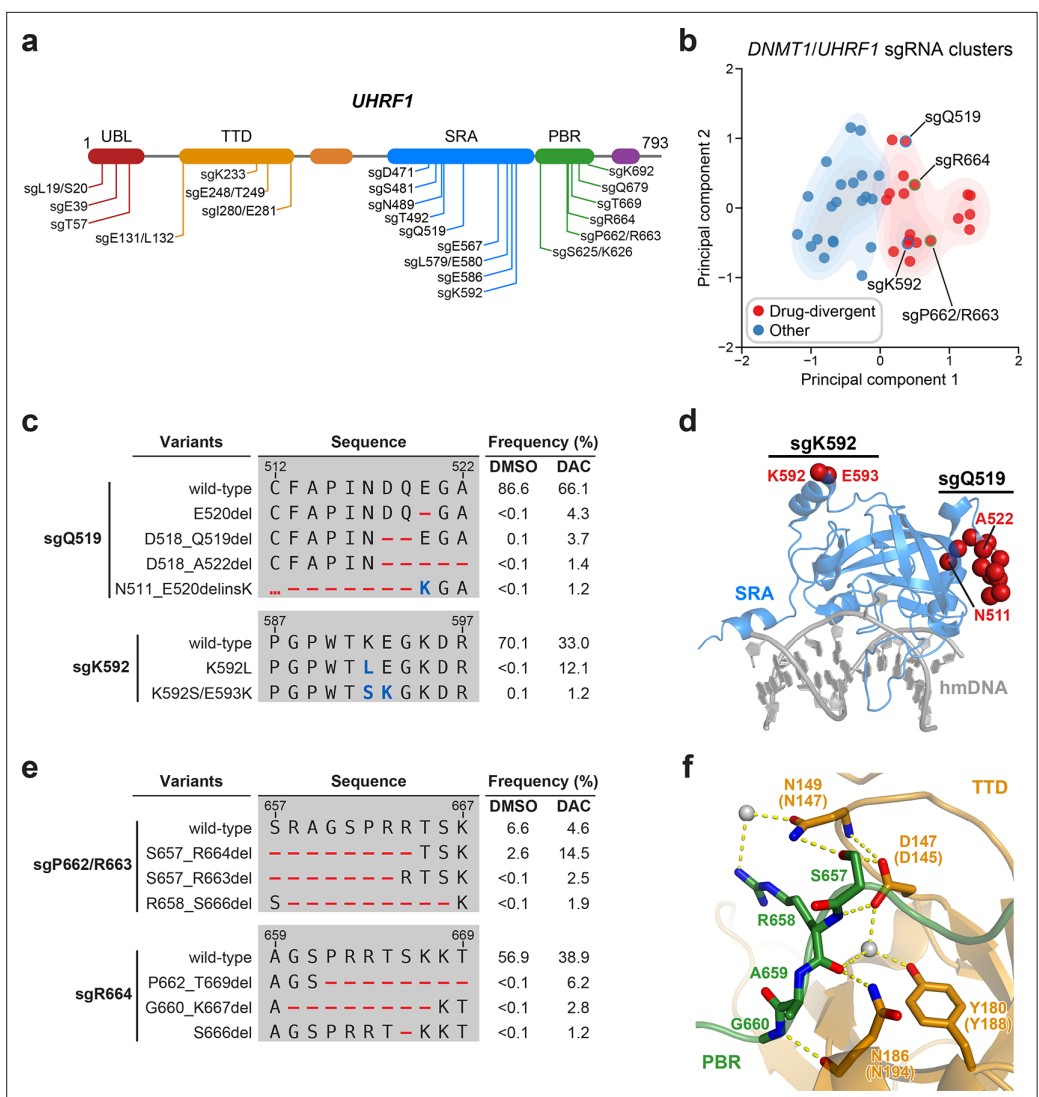

**Figure 5.** Mutational profiling analysis of individual single-guide RNAs (sgRNAs) nominates *UHRF1*-targeting sgRNAs with predicted gain-of-function outcomes. (**a**) Schematic showing the amino acid positions on the *UHRF1* CDS targeted by selected *UHRF1*-targeting sgRNAs (*n* = 22). (**b**) Scatter plot showing *DNMT1*- and *UHRF1*-targeting sgRNAs (*n* = 40; 18 and 22 for *DNMT1* and *UHRF1*, respectively) projected onto principal component space after principal component analysis on their mutational profile features. The sgRNAs were partitioned using *k*-means clustering (*k* = 2) into two categories: 'drug-divergent' (*n* = 19, red) or 'other' (*n* = 21, blue). Contours depict a bivariate kernel density estimation for drug-divergent (red) and other (blue) sgRNAs. sgRNAs highlighted in (**c**) and (**e**) targeting the SRA (blue border) and PBR (green border) regions of UHRF1 are annotated.(**c**) Table showing the amino acid sequence alignment and observed frequencies (percentage of total reads) of the wild-type and top enriched in-frame variants observed in the SRA-targeting sgRNAs sgQ519 (top) and sgK592 (bottom) after 8 weeks of treatment with vehicle or 100 nM decitabine (DAC). In-frame variants were considered enriched if the observed frequency in DAC was ≥1% and the $\log_2$(fold-change) of the observed frequency in DAC versus vehicle treatment was ≥2. All enriched in-frame variants meeting these criteria are shown and ordered by their observed frequency in DAC treatment. Amino acid deletions are represented as red dashes and substitutions are highlighted in blue. Red ellipses are used to denote amino acid deletions that exceed the length of the shown sequence alignment. (**d**) Structural view of the human UHRF1 SRA domain (blue) bound to hemimethylated DNA (hmDNA, gray). Residues perturbed by enriched in-frame variants (from **c**) found in sgQ519 and sgK592 are highlighted as red spheres (PDB: 3CLZ). (**e**) Table showing the amino acid sequence alignment and observed frequencies (percentage of total reads) of the wild-type and top enriched in-frame variants observed in the PBR-targeting sgRNAs sgP662/R663 (top) and sgR664 (bottom) after 8 weeks of treatment with vehicle or 100 nM DAC. In-frame variants were considered enriched if the observed frequency in DAC was ≥1% and the $\log_2$(fold-change) of the

*Figure 5 continued on next page*

*Figure 5 continued*

observed frequency in DAC versus vehicle treatment was ≥2. All enriched in-frame variants meeting these criteria are shown and ordered by their observed frequency in DAC treatment. Amino acid deletions are represented as red dashes and substitutions are highlighted in blue. Red ellipses are used to denote amino acid deletions that exceed the length of the shown sequence alignment. (**f**) Structural view of the zebrafish UHRF1 TTD domain (gold) complexed to a human UHRF1 PBR peptide (green) showing the region (S657–G660) targeted by sgP662/R663 and sgR664 (from **e**). Key residues forming polar contacts (yellow) are highlighted as sticks and annotated. For TTD residue annotations, the upper and lower (in parentheses) text indicate the residue identity and position in zebrafish and human UHRF1, respectively. Water molecules are shown as gray spheres (PDB: 6B9M).

The online version of this article includes the following figure supplement(s) for figure 5:

**Figure supplement 1.** Mutational profiling analysis of individual single-guide RNAs (sgRNAs) nominates *UHRF1*-targeting sgRNAs with predicted gain-of-function outcomes.

---

enrichment of multiple in-frame variants in DAC versus vehicle treatment for both sgRNAs (*Figure 5c*). By contrast, the top 9 most prevalent mutations in sgQ519 under vehicle treatment were loss-of-function (*Supplementary file 3*). Strikingly, the top 2 enriched in-frame variants for sgK592 in DAC were 3- and 4-nt substitutions, corresponding to K592L and K592S/E593K, respectively (*Figure 5c*). Because point mutations are uncommon Cas9 repair outcomes (*Shen et al., 2018*; *Hwang et al., 2020*) and given their low abundance in vehicle relative to DAC treatment (K592L, 12.1% in DAC versus <0.1% in vehicle; K592S/E593K, 1.2% in DAC versus 0.1% in vehicle), we speculate that these are rare editing outcomes that are highly selected for in DAC treatment. Our previous studies have also demonstrated that such strong selection pressures can indicate stringent mutational constraints imposed by structural or functional requirements of the local sequence (*Vinyard et al., 2019*; *Gosavi et al., 2022*; *Kwok et al., 2022*), suggesting that this region of the SRA may serve an important functional role.

Notably, enriched variants in sgQ519 and sgK592 perturb residues distal from the DNA-binding pocket of the SRA domain and the core structural elements forming the twisted β-barrel motif (*Figure 5d*). Previous studies investigating intramolecular interactions within UHRF1 with crosslinking mass spectrometry identified extensive contacts between residues proximal to sgQ519 and sgK592 (e.g., K524 and K595) and those in other UHRF1 domains (*DaRosa et al., 2018*; *Foster et al., 2018*). Accordingly, we speculate that these enriched variants are unlikely to disrupt DNA binding but may rather affect intra- or intermolecular interactions involving the SRA.

Beyond sgQ519 and sgK592, we observed several drug-divergent sgRNAs targeting the PBR region (sgP662/R663 and sgR664, *Figure 5e*), which mediates UHRF1 autoinhibition through an intramolecular interaction with the TTD domain (*Gelato et al., 2014*; *Fang et al., 2016*; *Gao et al., 2018*). This TTD–PBR interaction maintains a 'closed' conformation that prevents H3K9me3 binding and recruitment to chromatin. Disrupting this interaction, such as by SRA-mediated DNA binding, drives UHRF1 into an 'open' conformation that promotes H3K9me3 recognition and chromatin association. The top enriched in-frame variants induced by PBR-targeting sgRNAs perturb residues S657–G660, which are partially resolved in a structure of the zebrafish UHRF1 TTD complexed with human UHRF1 PBR peptide (*Figure 5f*). These residues form extensive contacts with the TTD (human UHRF1 residues D145, N147, Y188, and N194), suggesting that these mutations likely disrupt the TTD–PBR interaction and promote the open conformation of UHRF1. Supporting this notion, we observed that sgK233, a drug-divergent sgRNA targeting the TTD, also generated DAC-enriched in-frame mutations disrupting key residues on the other side of the TTD–PBR interface (*Figure 5—figure supplement 1h, i*). Taken together, our results suggest that disrupting the TTD–PBR interaction and relieving UHRF1 autoinhibition may confer a selective advantage to DAC. While further biochemical and cellular characterization are required to experimentally validate these potential perturbations to UHRF1 autoinhibition, our approach demonstrates how genotype-level mutational profiling of individual sgRNAs can afford valuable insight for nominating putative gain-of-function mutations for deeper mechanistic follow-up studies, especially in the absence of extensive structural data.

## Discussion

Despite significant advances, the discovery of allosteric mechanisms remains challenging. Here, we performed activity-based CRISPR scanning with the mechanistic inhibitor DAC to nominate multiple allosteric mechanisms regulating DNMT1 and UHRF1 function. Our study serves as an instructive framework that demonstrates how CRISPR scanning can be expanded beyond drug mechanism of action studies to identify regulatory sites in the direct drug target and even protein complex partners.

This study presents several key innovations to the CRISPR scanning methodology. First, we demonstrate how activity-based mechanistic inhibitors can enable rational screening of sophisticated phenotypes. DAC's unique properties as an activity-based inhibitor—being nearly identical to DNMT1's native substrate—precludes the enrichment of mutations disrupting drug binding as they would likely also abrogate DNMT1 activity, which is essential for cell survival. Indeed, our results support the idea that such constraints drive the enrichment of mutations that operate through alternative resistance mechanisms (e.g., by enhancing DNMT1 activity) as we not only recapitulated mutations targeting known autoinhibitory mechanisms in DNMT1 and UHRF1, but also discovered novel hypermorphic mutations in the uncharacterized BAH2 domain of DNMT1. Nevertheless, we cannot rule out other independent mechanisms by which these mutations might contribute to DAC resistance in cells. By contrast, the top hit in our CRISPR scanning screen of *DNMT1* and *UHRF1* with the reversible DNMT1 inhibitor GSK3484862 was sgH1507, which targets a key residue in the DNMT1 catalytic domain that is critical for drug binding. Although we have previously observed the enrichment of distal resistance mutations using non-covalent reversible inhibitors (*Vinyard et al., 2019*; *Gosavi et al., 2022*; *Kwok et al., 2022*), the distinct sgRNA enrichment profiles between DAC and GSK3484862 treatment suggest that activity-based probes can be leveraged to exert differential selection pressures that further predispose CRISPR scanning screens toward the enrichment of distal mutations. In this regard, elucidating the mechanistic details underlying how these distal DNMT1 mutations confer resistance to DAC is an avenue for future studies.

Second, our study significantly improves and expands the analysis toolkit that enables CRISPR scanning to identify putative functional hotspots. Our 1D and 3D sgRNA clustering analyses clearly implicate the DNMT1 autoinhibitory interface as a significantly enriched region of interest upon DAC treatment. Thus, our work emphasizes how an integrative approach incorporating orthogonal data, such as structural information, can nominate functional regions and provide mechanistic insights underlying their enrichment. We expect that improvements in computational approaches and the incorporation of other data (e.g., evolutionary conservation, human genetic variation, structural predictions; *Jumper et al., 2021*) may further increase the power of CRISPR scanning.

Although clustering of screen-level data is effective at nominating sgRNAs and putative hotspots for further validation, our findings demonstrate how the mutational profiles of individual sgRNAs at genotype-level resolution can uncover diverse responses to drug treatment that cannot be observed with screen-level enrichment scores. By clustering individual sgRNAs using their mutational profiles, we show that DNMT1 clusters K1 and K2 largely recapitulate the 3D structure-derived clusters with notable exceptions, such as the BAH2-targeting sgRNAs in cluster 2. Through an in-depth comparative analysis, we demonstrate that cluster K1 and K2 sgRNAs exhibit unique mutational signatures upon DAC treatment and identify defining characteristics of cluster K1 sgRNAs, such as the enrichment of in-frame mutations, that may be predictive of functional consequences. To validate our approach, we biochemically characterized a subset of these enriched BAH2 variants and show that these mutations are gain-of-function and enhance DNMT1 activity. Although overactive in vitro, these hypermorphic *DNMT1* mutations require further characterization in cells, such as by genomic DNA methylation profiling, to mechanistically understand how they confer resistance to DAC. Nevertheless, our analysis collectively demonstrates that the greater resolution of genotype-level data can reveal significant mutational heterogeneity across enriched sgRNAs, and that their mutational signatures can be exploited as a heuristic to nominate those with unique functional outcomes.

Notably, our study uses Cas nuclease as the editing modality, which is biased toward the formation of insertion/deletion (indel) mutations. While lacking precision, indel formation was critical for our studies, where (1) the stronger effect sizes afforded by larger perturbations were likely required to identify partial resistance mechanisms and (2) the diversity of editing outcomes enabled our genotype-level analysis and subsequent mechanistic insight. However, smaller, more precise perturbations such as point mutations may be more appropriate for other biological targets. In such cases,

our activity-based approach, which is primarily focused on the selection modality (i.e., mechanistic inhibitors), can also accommodate alternative editing technologies such as base editing or prime editing (*Anzalone et al., 2019*; *Anzalone et al., 2020*; *Hanna et al., 2021*; *Lue et al., 2023*) to assess distinct mutational perturbations. Indeed, we envision that integration of diverse editing modalities will further increase the utility of our activity-based approach.

Finally, we showcase how our approaches can be generalized to nominate sgRNAs that generate functional protein variants, especially in the absence of extensive structural information. We applied our mutational analysis methodology to nominate potential sites within UHRF1, a poorly characterized partner of DNMT1, that modulate DNMT1 function. Using a combined dataset of *UHRF1*- and *DNMT1*-targeting sgRNAs, we defined a set of 'drug-divergent' sgRNAs with similar characteristics as DNMT1 cluster K1 to identify putative gain-of-function variants in UHRF1. Strikingly, drug-divergent sgRNAs targeting *UHRF1* enriched for in-frame variants that perturb key residues on both sides of the TTD–PBR interface that mediates UHRF1 autoinhibition, in addition to uncharacterized regions of the SRA domain. Further study is necessary not only to experimentally validate the putative perturbations to UHRF1 autoinhibition described here, but also to contextualize how these UHRF1 mutations may influence DNMT1 activity, especially given the complex interplay between UHRF1 and DNMT1 required to coordinate and regulate DNA methylation maintenance (*Xie and Qian, 2018*; *Bronner et al., 2019*). Nonetheless, our findings outline how evaluating enriched sgRNAs at the genotype level can nominate variants for further functional validation.

To the best of our knowledge, there are no known *DNMT1* mutations reported to confer resistance to DAC or its structural analog 5-azacytidine (AZA) in the clinic. Rather, secondary resistance to these DNA hypomethylating agents develops primarily through adaptive responses in drug metabolism genes (e.g., *DCK* and *UCK2*) that disrupt the processing required for DAC/AZA incorporation into DNA, or drug transporter genes (e.g., *SLC29A1* and *SLC29A2*) that disrupt cellular uptake of DAC/AZA (*Qin et al., 2009*; *Gruber et al., 2020*; *Gu et al., 2021*). Consistent with this, a recent study of DAC/AZA resistance mechanisms using genome-wide CRISPR-knockout screens observed overwhelming enrichment of sgRNAs exclusively targeting *DCK*, *UCK2*, and *SLC29A1* (*Gruber et al., 2020*). Compared to inducing hypermorphic *DNMT1* mutations or reducing *DNMT1* copy number, the ease by which drug metabolism or uptake pathways can be altered in response to DAC/AZA treatment may explain why *DNMT1* mutations are not observed in patients with acquired resistance. The identification of hypermorphic *DNMT1* mutations and other putative gain-of-function variants reported here is presumably a unique result of our selective Cas9-mediated mutagenesis of *DNMT1* and *UHRF1*. These observations further underscore the utility of targeted CRISPR-mutagenesis approaches for interrogating protein function, as such variants are unlikely to arise in an unbiased screen. Thus, our approach can enable the discovery of resistance mutations that can afford valuable mechanistic insight into protein function but may otherwise be inaccessible due to the existence of alternative resistance mechanisms.

Altogether, here we demonstrate how activity-based CRISPR scanning can be leveraged to nominate allosteric regulatory mechanisms, using DNMT1 and UHRF1 as instructive paradigms. Through an array of genetic, biochemical, and computational approaches, we illustrate how integrative analyses can offer mechanistic insights at increasing levels of resolution. In summary, our study establishes a framework for applying CRISPR scanning to systematically identify allosteric mechanisms and other complex phenotypes across various protein targets.

## Materials and methods
### Chemical reagents
Compounds were stored at −80°C in 100% dimethyl sulfoxide (DMSO; Sigma-Aldrich). The vehicle condition represents 0.1% DMSO treatment. Decitabine (DAC) was purchased from Selleck Chemicals (≥99% purity by high-performance liquid chromatography [HPLC]). GSK3484862 (GSKi) was purchased from ChemieTek (≥99% purity by HPLC).

### Cell culture
K562 cells were obtained from ATCC (cat. #CCL-243, RRID: CVCL_0004). HEK293T cells were a gift from B.E. Bernstein (Massachusetts General Hospital). All cell lines were cultured in a humidified 5% $CO_2$ incubator at 37°C and routinely tested for mycoplasma (Sigma-Aldrich). All media were supplemented

with 100 U ml$^{-1}$ penicillin and 100 µg ml$^{-1}$ streptomycin (Gibco) and fetal bovine serum (FBS, Peak Serum). K562 cells were cultured in RPMI-1640 (Gibco) supplemented with 10% FBS. HEK293T cells were cultured in Dulbecco's Modified Eagle Medium (DMEM; Gibco) supplemented with 10% FBS.

## Lentivirus production and transduction

To produce lentivirus, transfer plasmids were co-transfected with *GAG/POL* and *VSVG* plasmids into HEK293T cells using FuGENE HD (Promega) and the medium was exchanged 6–8 hr after transfection. Viral supernatant was collected 48–60 hr after transfection, filtered (0.45 µm), and stored at −80°C until use. K562 cells were transduced by spinfection at 1800 × *g* for 90 min at 37°C with 8 µg ml$^{-1}$ polybrene (Santa Cruz Biotechnology) and selected with 2 µg ml$^{-1}$ puromycin (Gibco) starting 48 hr post-transduction.

## Plasmid construction

For individual sgRNA validation experiments, single-stranded oligonucleotides encoding the sgRNA (Sigma-Aldrich) were annealed and cloned into lentiCRISPRv2 (a gift from F. Zhang, Addgene #52961) using Golden Gate cloning with FastDigest Esp3I (Thermo Fisher Scientific) and T4 Ligase (New England Biolabs). For protein cellular stability and degradation experiments, full-length human DNMT1 was cloned into the Artichoke reporter plasmid (a gift from B. Ebert, Addgene #73320) using Gibson Assembly with NEBuilder HiFi (New England Biolabs). For DNMT1 knockdown and overexpression experiments, the *DNMT1* 3′ UTR shRNA (TRCN0000232751) lentiviral expression vector was obtained from Sigma-Aldrich and full-length human DNMT1 was cloned with an N-terminal HA-tag into a lentiviral expression vector using Gibson Assembly. For bacterial expression constructs, truncated human DNMT1 (residues 351–1616) was cloned into pET15b containing an N-terminal His$_6$-tag and TEV protease cleavage site using Gibson Assembly. The wild-type human *DNMT1* CDS was subcloned from pcDNA3.1–HA–DNMT1, a gift from D. Tenen (Harvard Medical School) and mutations were introduced by modifying the primers used to amplify the *DNMT1* CDS for Gibson Assembly.

## Cell line generation and validation

For DNMT1 knockdown and overexpression experiments, lentivirus for the *DNMT1* shRNA and HA–DNMT1 overexpression constructs was produced as described above. Wild-type K562 cells were then co-transduced with the *DNMT1* shRNA construct lentivirus in addition to the appropriate DNMT1 overexpression construct lentivirus as described above and treated with 2 µg ml$^{-1}$ puromycin and 5 µg ml$^{-1}$ blasticidin (Gibco) for 7 days to select for cells transduced with both constructs. Additionally, a *DNMT1* shRNA construct-only transduction was performed and selected with 2 µg ml$^{-1}$ puromycin as a control for shRNA knockdown of endogenous *DNMT1*. After selection, shRNA knockdown of endogenous *DNMT1* and expression of the exogenous HA–DNMT1 construct was validated by immunoblotting analysis. To generate the *DNMT1*-mutant clonal cell lines, the surviving cells after 8 weeks of DAC treatment in the CRISPR scanning experiment were sorted as single cells in 96-well plates (Corning) and then expanded before isolating genomic DNA. The sgRNA expression cassette was amplified from the genomic DNA by polymerase chain reaction (PCR) and Sanger sequenced to determine the protospacer sequence and its corresponding genomic locus. The target region was then amplified using genomic primers with Illumina adapters and sequenced on an Illumina MiSeq using 300-cycle, single-end reads. Sequencing data were processed with CRISPResso2 (*Clement et al., 2019*) to determine the genotypes of the clonal lines.

## Cell growth assays

For cell growth assays with GSKi treatment, wild-type K562 cells were seeded in 24-well plates (Corning) at a density of 2 × 10$^5$ cells ml$^{-1}$ and treated with GSKi or vehicle in triplicate. For cell growth assays performed with clonal cell lines or cell lines transduced with DNMT1 shRNA and overexpression vectors, wild-type and mutant (clonal or transduced) K562 cells were seeded in 96-well plates at a density of 1 × 10$^5$ cells ml$^{-1}$ and treated with DAC or vehicle in triplicate. In all cell growth assays, the cells were passaged with fresh media containing drug (GSKi or DAC) or vehicle on day 3 and cell viability was assessed at day 7 by flow cytometry after viability staining with Helix NP NIR dye (BioLegend). Relative growth was then calculated by comparing cell viability at the indicated treatment

condition to vehicle treatment. Dose–response curves were determined by fitting a 4-parameter logistic regression to the data using the SciPy (*Virtanen et al., 2020*) package (v.1.7.1).

## Cellular protein stability and degradation assays

Lentivirus was produced for wild-type and mutant DNMT1–EGFP–IRES–mCherry (Artichoke) reporter constructs as described above. Wild-type K562 cells were then transduced with the appropriate lentivirus and selected with puromycin for 3 days as described above. The selected cells for each construct were then split into two pools and treated with vehicle or 100 nM DAC for 3 days in triplicate, after which EGFP and mCherry fluorescence were measured on a NovoCyte 3000RYB flow cytometer (Agilent). The geometric mean of the ratio of EGFP to mCherry fluorescence was calculated for mCherry-positive cells (see *Figure 3—figure supplement 1f*) in each sample using the NovoExpress software (Agilent). To assess cellular stability, the EGFP/mCherry ratios of the mutant constructs in vehicle treatment were normalized to the wild-type EGFP/mCherry ratio in vehicle treatment. To assess degradation, the EGFP/mCherry ratios in DAC treatment for each construct were normalized to their respective EGFP/mCherry ratios in vehicle treatment.

## Immunoblotting

Cells were harvested, washed three times with cold phosphate-buffered saline (Corning), and lysed in radio-immunoprecipitation assay (RIPA) buffer (Boston BioProducts) supplemented with 1× Halt Protease Inhibitor Cocktail and 5 mM ethylenediaminetetraacetic acid (EDTA; Thermo Fisher Scientific) on ice for 30 min. Lysates were clarified by centrifugation and total protein concentration in clarified lysates was determined using the BCA Protein Assay Kit (Thermo Fisher Scientific) prior to preparing samples for sodium dodecyl sulfate–polyacrylamide gel electrophoresis (SDS–PAGE). Immunoblotting was performed according to standard procedures using the following primary antibodies: anti-DNMT1 (clone D63A6, Cell Signaling Technology, cat. #5032, RRID: AB_10548197, 1:1000), anti-GAPDH (clone 0411, Santa Cruz Biotechnology, cat. #sc-47724, RRID: AB_627678, 1:2000), anti-HA (clone C29F4, Cell Signaling Technology, cat. #3724, RRID: AB_1549585, 1:1000), and anti-LMNB1 (Abcam, cat. #ab16048, RRID: AB_443298, 1:2000).

## Protein expression and purification

Wild-type and mutant human DNMT1$_{351–1616}$ bacterial expression constructs were cloned as described above. Recombinant DNMT1 expression and purification were performed according to published protocol (*Dolen et al., 2019*) with some modifications. DNMT1 expression constructs were transformed and expressed recombinantly in *Escherichia coli* Rosetta2(DE3)pLysS cells (Novagen). Freshly transformed cells were grown in LB broth supplemented with ampicillin and chloramphenicol at 37°C to an OD$_{600}$ of 0.6, after which the cells were cooled on ice and induced with 0.4 mM isopropyl-β-ᴅ-thiogalactoside (Research Products International) at 16°C overnight. Cells were harvested, pelleted by centrifugation, and stored at −80°C until use. Cells were resuspended in lysis buffer containing 25 mM Tris–HCl (pH 7.5), 500 mM NaCl, 4 mM β-mercaptoethanol (BME), 5% glycerol, 3 U ml$^{-1}$ DNase I (New England Biolabs), and 1× cOmplete EDTA-free protease inhibitor cocktail (Roche), lysed by sonication, and clarified by centrifugation. Clarified lysate was incubated with His60 Ni Superflow resin (Takara Bio) for 1 hr at 4°C and then washed with buffer containing 20 mM Tris–HCl (pH 7.5), 500 mM NaCl, 4 mM BME, 5% glycerol, and 20 mM imidazole. Protein was eluted with buffer containing 20 mM Tris–HCl (pH 7.5), 500 mM NaCl, 4 mM BME, 5% glycerol, and 400 mM imidazole. The eluate was diluted with an equal volume of buffer containing 20 mM sodium phosphate (pH 7.5), 2 mM dithiothreitol (DTT), and 5% glycerol and then further purified on a HiTrap Heparin HP column (Cytiva) using a linear gradient of 0.25–1.5 M NaCl in buffer containing 20 mM sodium phosphate (pH 7.5), 2 mM DTT, and 5% glycerol. Fractions containing DNMT1 were pooled and concentrated with Amicon Ultra 30 kDa centrifugal filters (EMD Millipore) and purity was verified by SDS–PAGE. Purified proteins were quantified by absorbance at 280 nm and stored in 40% glycerol at −80°C until use.

## DNMT1 enzymatic activity assays

DNMT1 enzymatic activity was measured using the MTase-Glo Methyltransferase Assay (Promega) with recombinant DNMT1$_{351–1616}$ and a 14-bp oligonucleotide substrate (Integrated DNA Technologies) containing a single hemimethylated CpG site. Methyltransferase reactions were prepared with

800 nM recombinant DNMT1$_{351–1616}$, 10 µM hemimethylated DNA substrate, 10 µM S-adenosyl-L-methionine, and 1× MTase Glo Reagent in reaction buffer (20 mM Tris–HCl [pH 8.0], 50 mM NaCl, 3 mM MgCl$_2$, 1 mM DTT, 1 mM EDTA, 0.1 mg ml$^{-1}$ BSA) and incubated for 90 min at 30°C. For temperature-dependent enzyme activity assays, additional reactions were prepared and incubated at 23, 30, and 37°C. Following incubation, an equal volume of MTase-Glo Detection Solution was added to each reaction and further incubated for 30 min at room temperature. Reactions were then plated in technical triplicate in 20 µl volumes into a white, opaque 384-well plate (Corning) and endpoint luminescence was measured using a SpectraMax i3x plate reader (Molecular Devices). To account for baseline luminescence, raw luminescence values were corrected by subtracting the average luminescence values of control reactions prepared without DNMT1. All activity assays were independently conducted at least twice.

## Pooled sgRNA library cloning and CRISPR scanning experiments

The pooled sgRNA tiling library was designed with CRISPOR (*Concordet and Haeussler, 2018*) using the following criteria: (1) the 20-nt protospacer sequence must be upstream of an NGG PAM, (2) the predicted cleavage site falls within the coding sequence of *DNMT1* (*NP_001370.1*) or *UHRF1* (*NP_001041666.1*), and (3) the sgRNA must have an off-target score (MIT Specificity Score) greater than 20. All sgRNAs meeting, these criteria were synthesized as an oligonucleotide pool (Twist Biosciences) and their sequences are listed in *Supplementary file 1*. The sgRNA oligo pool was amplified, cloned into lentiCRISPRv2, and sequenced to confirm sgRNA representation as previously described (*Ngan et al., 2022*; *Joung et al., 2017*; *Canver et al., 2018*). Lentivirus carrying the resulting pooled sgRNA tiling library was produced as described above and titered according to published procedure (*Ngan et al., 2022*; *Joung et al., 2017*; *Canver et al., 2018*).

For CRISPR scanning experiments, K562 cells (40 × 10$^6$) were transduced at a multiplicity of infection <0.3 and subsequently selected with puromycin for 4 days. Cells were then split into pools and treated with DAC (100 nM for 5 weeks and then 1 µM for 3 weeks) or vehicle in triplicate. For the GSKi CRISPR scanning experiment, cells were treated with GSKi (1 µM for 3 weeks followed by 5 µM for 3 weeks) or vehicle in triplicate. The cells were passaged every 3–4 days at a seeding density of 0.1–0.2 × 10$^6$ cells ml$^{-1}$ into fresh media containing drug or vehicle. Genomic DNA was isolated using the QIAamp DNA Blood Mini Kit (Qiagen). To measure the sgRNA composition of the population, the sgRNA expression cassette was PCR amplified using barcoded primers, purified, and sequenced as previously described (*Vinyard et al., 2019*; *Ngan et al., 2022*; *Joung et al., 2017*; *Canver et al., 2018*). All samples were sequenced on a MiSeq (Illumina) using 150-cycle, single-end reads. Sufficient coverage of the sgRNA library was maintained in accordance with published recommendations (*Ngan et al., 2022*; *Joung et al., 2017*; *Canver et al., 2018*).

## CRISPR scanning data analysis

All data processing and analysis were performed using Python v.3.8.3 (https://www.python.org/). Raw sequencing data were processed as previously described (*Vinyard et al., 2019*; *Gosavi et al., 2022*). In brief, reads were counted by identifying the 20-nt sequence downstream of the 'CGAA ACACCG' prefix and mapped against a reference file containing all library sgRNA sequences with no mismatch allowance. sgRNAs with zero reads in the plasmid library were excluded from the analysis. Read counts were then converted to reads per million, increased by a pseudocount of 1, log$_2$-transformed, and then normalized by subtracting the log$_2$-transformed sgRNA counts in the plasmid library. Library-normalized scores were averaged across replicates for each condition and sgRNA 'resistance scores' were calculated by first subtracting the scores in vehicle treatment from their corresponding scores in DAC treatment, and then further normalized by subtracting the mean resistance score of the negative control sgRNAs from all sgRNAs. sgRNAs were classified as 'enriched' if their resistance scores were greater than the mean resistance score plus 2 standard deviations (SDs) of the negative control sgRNAs. sgRNAs were assigned to protein amino acid positions by using the genomic coordinates of their predicted cut sites in the *DNMT1* (*NP_001370.1*) or *UHRF1* (*NP_001041666.1*) coding sequences. sgRNAs were assigned to a single amino acid if the cut site fell within a codon or assigned to the two flanking amino acids if the cut site fell between codons.

## Linear clustering analysis

As proximal sgRNAs can exhibit significant variation due to sgRNA-specific factors (e.g., off-target activity, cutting efficiency), per-residue resistance scores were estimated with respect to local sgRNAs by using LOESS regression to fit the observed sgRNA resistance scores as a function of amino acid position. To estimate resistance scores for each amino acid in DNMT1, LOESS regression was performed using the 'lowess' function of the statsmodels package (v.0.12.1) in Python with a 100 AA sliding window ('frac = (100 AA/$L$)', where $L$ is the total length of the protein), and 'it = 0'. For amino acid positions that were not targeted by sgRNAs, resistance scores were interpolated by performing quadratic spline interpolation on the LOESS output scores using the 'interp1d' function of the SciPy (*Virtanen et al., 2020*) package (v.1.7.1).

To assess statistical significance of the resulting clusters, we simulated a null model of random sgRNA enrichment. sgRNA cut site positions were kept fixed while sgRNA resistance scores were randomly shuffled, and per-residue resistance scores were recalculated by performing LOESS regression and interpolation on the randomized sgRNA resistance scores for each of 10,000 permutations. Empirical p values were calculated for each amino acid by comparing its observed resistance score to the null distribution of random resistance scores. Empirical p values were adjusted using the Benjamini–Hochberg procedure to control the false discovery rate to ≤0.05. Finally, linear clusters were called by identifying all contiguous intervals of amino acids with adjusted p values ≤0.05.

## 3D spatial clustering analysis

To perform the 3D spatial clustering analysis, we first calculated PWES between pairs of sgRNAs. We employed a modified version of previously published procedures (*Vinyard et al., 2019*; *Kamburov et al., 2015*), using a scoring function involving (1) the pairwise score for a given pair of sgRNAs and (2) the Euclidean distance between their targeted residues. First, for all combinations of *DNMT1*-targeting sgRNAs $i$ and $j$, the pairwise score $pw_{i,j}$ was calculated using the following function:

$$pw_{i,j} = tanh\left(\frac{x_{i,j} - \bar{x}}{s_x}\right)$$

where $x_{i,j}$ is the sum of the sgRNA resistance scores for a given pair of sgRNAs $i$ and $j$ ($i \neq j$), while $\bar{x}$ and $s_x$ are the mean and SD, respectively, of the summed resistance scores for all pairwise combinations of *DNMT1*-targeting sgRNAs. The hyperbolic tangent function was used to scale pairwise scores in order to minimize the disproportionate influence of highly enriched or depleted (i.e., jackpotted) sgRNAs and normalize them into the interval of [−1,1].

Next, we determined the distances between all pairwise combinations of resolved amino acids in the structure of human DNMT1$_{351-1600}$ (PDB: 4WXX) by calculating the Euclidean distance between the centroids of the two residues using PyMOL (v.2.5.0, Schrödinger). We then isolated the subset of sgRNAs whose assigned amino acid positions were resolved in the structure. sgRNAs predicted to cut between residues were assigned to the even-numbered residue. Thus, the final PWES for all pairwise combinations of resolved sgRNAs $i$ and $j$ were calculated as follows:

$$PWES_{i,j} = pw_{i,j} \cdot e^{\frac{-d_{i,j}^2}{2t^2}}$$

where $d_{i,j}$ is the Euclidean distance between the targeted residues of sgRNAs $i$ and $j$ and $t = 16$. Hierarchical clustering was performed as previously described on the resultant pairwise PWES matrix to group sgRNAs by their PWES profiles (*Vinyard et al., 2019*).

To assess the significance of clusters 1 and 2, their sgRNA resistance scores were kept fixed while their targeted amino acid positions were randomly shuffled ($n = 10,000$) across the amino acids targeted by resolved sgRNAs to simulate a null distribution of PWES values with randomized spatial proximity. We then took the sum of the absolute values of PWES for all intra-cluster pairwise sgRNA combinations to calculate the 'summed PWES' score per cluster. Empirical p values were calculated by comparing the observed summed PWES score for a cluster to the simulated distribution of summed PWES scores.

## Individual sgRNA validation experiments and genotyping data analysis

Individual sgRNAs selected for further validation were cloned into lentiCRISPRv2 as described above. The sequences for the sgRNAs used in these experiments are listed in *Supplementary file 4*. Lentivirus was produced separately for each sgRNA construct as described above and K562 cells were transduced and selected with puromycin for 5 days. For the individual sgRNA timecourse experiment, the selected cells were cultured in standard growth conditions and passaged every 3–4 days, and cells were harvested at the indicated timepoints. For all other individual sgRNA experiments, the selected cells in each sgRNA transduction condition were then split into two pools and treated with vehicle or 100 nM DAC for 8 weeks before harvesting. Genomic DNA was isolated from harvested cells using QuickExtract DNA Extraction Solution (Biosearch Technologies) and used to prepare libraries for next-generation sequencing as described previously (*Joung et al., 2017*). Briefly, the genomic region surrounding the predicted cut site of each sgRNA was first PCR amplified using genomic primers with Illumina adapters, followed by a second round of PCR to attach barcodes to the final amplicons. The final amplicons were then gel-purified using the Zymoclean Gel DNA Recovery Kit (Zymo Research), pooled, and sequenced on an Illumina MiSeq using 300-cycle, single-end reads. Primer sequences are provided in *Supplementary file 4*.

To identify genomic variants and quantify allele frequencies, raw sequencing data were processed and aligned to *DNMT1* and *UHRF1* using CRISPResso2 (*Clement et al., 2019*) (v.2.0.40) with the following parameters: '-w 30 -q 10 –min_bp_quality_or_N 10 –exclude_bp_from_left 5 –exclude_bp_from_right 5 –plot_window_size 30'. CRISPResso2 allele frequency outputs were further processed with custom Python scripts to classify and characterize variants at the protein level for downstream analysis. Reads with no editing were classified as 'wild-type'. For all variants with mutations within the coding sequence, variants were classified as 'in-frame' if the net indel size was a multiple of three and 'frameshift' if not. Variants with mutations that span an intron–exon junction or otherwise disrupted canonical splice site positions (the 2 nt immediately flanking each exon) were classified as 'splice site disrupting'. In-frame variants were then further processed into their corresponding protein variants by performing global re-alignment to the reference CDS at the nucleotide level using a customized codon-based implementation of the Needleman–Wunsch algorithm using the 'PairwiseAligner' module of Biopython (*Cock et al., 2009*) (v.1.7.8), followed by trimming and translation. Translated in-frame variants were further classified as 'nonsense' if the mutation led to a premature stop codon or merged with 'wild-type' if the translation protein variant matched the reference protein sequence (e.g., silent mutations, SNPs). Finally, all variants identified as frameshift, splice site disrupting, or nonsense were further classified as 'loss-of-function'. After processing, the final allele tables for each sgRNA were filtered to only include variants with read frequencies ≥0.1% in either vehicle or DAC treatment and frequencies re-normalized to 100%. Processed variants and their read frequencies are supplied in *Supplementary file 2* and *Supplementary file 3*.

Editing outcome predictions for individual gRNAs were obtained using the inDelphi (*Shen et al., 2018*) web server (https://indelphi.giffordlab.mit.edu) in single mode with K562 as the cell type. As inDelphi does not consider intron-exon boundaries or translated protein products, inDelphi-predicted genotypes were also processed similarly as above in order to classify variants and accurately determine the predicted frequencies of in-frame versus loss-of-function mutations.

## Individual sgRNA mutational profile analysis and clustering

Processed and filtered allele tables were used to calculate the various metrics for the mutational profile analysis of individual sgRNAs. Absolute variant frequencies were calculated by dividing the reads assigned to a particular variant by the total number of reads. Relative variant frequencies were calculated with respect to the total number of reads assigned to edited (i.e., in-frame or loss-of-function, wild-type excluded) variants. $\log_2$(fold-change) metrics for wild-type, in-frame, and loss-of-function mutation types were calculated as the absolute frequency of the mutation type in DAC divided by the absolute frequency in vehicle, followed by $\log_2$-transformation.

Log-odds were calculated for the two binary outcomes of edited versus wild-type (edited/WT) and in-frame versus loss-of-function (IF/LOF) as follows:

$$\log_2\left(\text{odds}_{\text{edited/WT}}\right) = \log_2\left(\frac{(f_{\text{abs}}(\text{IF}) + f_{\text{abs}}(\text{LOF}))}{f_{\text{abs}}(\text{WT})}\right)$$

$$\log_2\left(\text{odds}_{\text{IF/LOF}}\right) = \log_2\left(\frac{f_{\text{rel}}(\text{IF})}{f_{\text{rel}}(\text{LOF})}\right)$$

$$f_{\text{abs}}(\text{IF}) + f_{\text{abs}}(\text{LOF}) + f_{\text{abs}}(\text{WT}) = 1$$

$$f_{\text{rel}}(\text{IF}) + f_{\text{rel}}(\text{LOF}) = 1$$

where $f_{abs}$ and $f_{rel}$ refer to the absolute and relative frequency, respectively, of the mutation type. Log-odds for edited versus wild-type alleles were calculated using absolute frequencies, whereas log-odds for in-frame versus loss-of-function alleles were calculated using relative frequencies. Log-odds ratios comparing DAC to vehicle or inDelphi were calculated by subtracting the log-odds in vehicle or inDelphi from the log-odds value in DAC.

Pearson correlations were calculated on the absolute variant frequencies in DAC and vehicle treatments using the 'stats.pearsonr' function of SciPy (*Virtanen et al., 2020*) Gini coefficients were calculated as follows:

$$\text{Gini coefficient} = 2 \cdot AUC - 1$$

where *AUC* is the area under the curve of the empirical cumulative distribution function of allele frequencies. Gini coefficients were calculated with respect to all alleles using absolute frequencies as well as edited alleles using relative frequencies.

To assess the similarity of mutational profiles across treatment conditions, we used the symmetric KL divergence, which is calculated for two probability distributions *P* and *Q* (i.e., allele frequency distributions in DAC and vehicle, respectively) as the sum of the standard KL divergences of *P* from *Q* and *Q* from *P* as follows:

$$KL = D_{KL}\left(P\|Q\right) + D_{KL}\left(Q\|P\right)$$

where the standard KL divergence of *P* from *Q* is calculated as:

$$D_{KL}\left(P\|Q\right) = \sum_i P_i log\left(\frac{P_i}{Q_i}\right)$$

where *i* indexes the alleles found in each sgRNA, and $P_i$ and $Q_i$ are the frequencies of allele *i* in samples *P* and *Q* (i.e., DAC and vehicle treatments). To avoid division by zero, a pseudocount of 0.01% was added to all allele frequencies.

Subsequent data preprocessing, PCA, and *k*-means clustering were performed in Python using the scikit-learn package (v.0.24.2) (*Pedregosa, 2012*). Feature input data were first preprocessed by independently applying a rank-based quantile transformation on each feature using the 'QuantileTransformer' function. PCA was performed on the transformed dataset using the 'sklearn.decomposition.PCA' function with 'n_components = 10'. To cluster sgRNAs based on their mutational profile features, *k*-means clustering was performed on the resultant PCA matrix using the 'sklearn.cluster.KMeans' function with 'n_clusters = 2' and 'n_init = 1000'. To verify the fidelity of the clusters, *k*-means clustering was performed 1000 times and the most common outcome was used for the final *k*-means cluster assignments.

## Statistical methods and replication

Statistical parameters including the exact value and definition of *n*, the definition of center, dispersion, precision measures (mean ± SD or standard error of the mean), and statistical significance are reported in figures and figure legends. All statistical tests were performed as two-sided tests using the SciPy (*Virtanen et al., 2020*) package (v1.7.1). All experiments were performed at least twice except for those involving next-generation sequencing, which were conducted once.

## Code availability

Custom code used in this study is available at https://github.com/liaulab/DNMT1_eLife_2022 (*Liau Lab, 2028*; copy archived at swh:1:rev:18d3fc1571a461096a7baaf4251439ce39123d65).

## Acknowledgements

We thank members of the Liau Lab for helpful discussions and comments on the manuscript, in particular A Siegenfeld and A Waterbury. We thank K Zhao and P Randolph for assistance with next-generation sequencing. We thank E Dolen and R Switzer for advice regarding protein expression and purification. We thank R Puram for helpful discussion on clinical DNMT1 mutations and DAC resistance mechanisms. We thank Z Niziolek and J Nelson of the Bauer Core Facility for assistance with single-cell sorting.

## Additional information

### Competing interests

Brian B Liau: holds sponsored research projects with Eisai and AstraZeneca, is a scientific consultant for Imago BioSciences and Exo Therapeutics, and is a shareholder and member of the scientific advisory board of Light Horse Therapeutics. The other authors declare that no competing interests exist.

### Funding

| Funder | Grant reference number | Author |
| --- | --- | --- |
| National Science Foundation | DGE1745303 | Kevin Chun-Ho Ngan Nicholas Z Lue Emma M Garcia |
| National Institute of General Medical Sciences | 1DP2GM137494 | Brian B Liau |
| Damon Runyon Cancer Research Foundation | Damon Runyon-Rachleff Innovation Award | Brian B Liau |
| Harvard University | Landry Cancer Biology Fellowship | Emma M Garcia |
| Harvard University | Herchel Smith Graduate Fellowship | Ceejay Lee |
| Harvard University | Startup Funding | Brian B Liau |
| Health Resources in Action | Charles A. King Trust Postdoctoral Research Fellowship | Hui Si Kwok |

The funders had no role in study design, data collection, and interpretation, or the decision to submit the work for publication.

### Author contributions

Kevin Chun-Ho Ngan, Conceptualization, Resources, Data curation, Software, Formal analysis, Supervision, Validation, Investigation, Visualization, Methodology, Writing – original draft, Project administration, Writing – review and editing; Samuel M Hoenig, Validation, Investigation, Writing – review and editing; Hui Si Kwok, Nicholas Z Lue, Emma M Garcia, Validation, Investigation, Methodology, Writing – review and editing; Pallavi M Gosavi, Supervision, Methodology, Writing – review and editing; David A Tanner, Formal analysis, Validation, Investigation, Visualization, Writing – review and editing; Ceejay Lee, Resources, Software, Writing – review and editing; Brian B Liau, Conceptualization, Supervision, Funding acquisition, Methodology, Writing – original draft, Project administration, Writing – review and editing

### Author ORCIDs

Kevin Chun-Ho Ngan http://orcid.org/0000-0001-8067-3472
Hui Si Kwok http://orcid.org/0000-0002-2858-8876
Nicholas Z Lue http://orcid.org/0000-0002-4236-9127
Emma M Garcia http://orcid.org/0000-0001-5111-1622
Brian B Liau http://orcid.org/0000-0002-2985-462X

### Decision letter and Author response

Decision letter https://doi.org/10.7554/eLife.80640.sa1

Author response https://doi.org/10.7554/eLife.80640.sa2

## Additional files

### Supplementary files
- Supplementary file 1. *DNMT1* and *UHRF1* sgRNA sequences and CRISPR scanning data.
- Supplementary file 2. Variant frequencies for *DNMT1*-targeting sgRNAs.
- Supplementary file 3. Variant frequencies for *UHRF1*-targeting sgRNAs.
- Supplementary file 4. Oligonucleotide and PCR primer sequences.
- MDAR checklist

### Data availability
All data generated or analyzed during this study are included in the manuscript and supplementary files; Source Data for the CRISPR scanning experiments and individual sgRNA validation experiments are provided in Supplementary Files 1–3. The sequences of primers and oligonucleotides used in this study are provided in Supplementary File 4. Custom scripts used to analyze the data are available at https://github.com/liaulab/DNMT1_eLife_2022, (copy archived at swh:1:rev:18d3fc1571a461096a7baaf4251439ce39123d65).

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
