## [Editor Report]

This manuscript describes the use of genome editing screens to identify mechanisms underlying resistance to the hypomethylating anti-cancer agent decitabine, an activity-based inhibitor of the DNA methyltransferase DNMT1. A specific focus is given to the development of tools and approaches to identify allosteric mechanisms of resistance that emerge, including those that appear to act as gain-of-function mutations in DNMT1 and its interacting partner UHRF1. These findings showcase the power of large-scale genome editing for uncovering novel resistance mechanisms and investigating protein allostery.

---

## [Decision Letter]

**Decision letter after peer review:**

Thank you for submitting your article "Activity-based CRISPR Scanning Uncovers Allostery in DNA Methylation Maintenance Machinery" for consideration by *eLife*. Your article has been reviewed by 2 peer reviewers, and the evaluation has been overseen by a Reviewing Editor and Jessica Tyler as the Senior Editor. The reviewers have opted to remain anonymous.

Essential revisions:

1) To better support the claim that "activity-based CRISPR scanning" can enrich for resistance mutations in allosteric sites versus orthosteric sites, the reviewers find it would be important to perform a head-to-head comparison with a standard reversible DNMT1 inhibitor. Alternatively, since the reviewers thought that the current screens were of sufficient quality and general interest, the authors could instead elect to tone down the claims about the special utility of activity-based CRISPR scanning in the context of enriching allosteric resistance mechanisms.

2) The Reviewers felt that the manuscript would be strengthened by further characterization of a subset of the resistance mutants to better understand their mechanism of action. Much seems to be inferred from the location of the resistance mutants without experimental follow-up, and these conclusions were viewed as being somewhat speculative at this stage. For instance, the clustering of the apparent LOF mutations around the active site was intriguing, but also perplexing – if reflecting simply a gene dosage effect, why wouldn't such LOF mutations be more broadly distributed throughout the protein? Could a different (and possibly more interesting) mechanism of resistance underpin this cluster of catalytic site mutations? Or, more generally, can the clustering of mutations in the same region of the protein be safely concluded to reflect a shared mechanism for these mutations?

*Reviewer #1 (Recommendations for the authors):*

I think the DNMT1 screening data is sufficiently interesting and rigorous to be appropriate for publication in *eLife*, but I think one of two changes should be considered. Either a head-to-head comparison should be made to support the claim that this "activity-based CRISPR scanning" yields a preference for allosteric site discovery over "CRISPR suppressor scanning" or the authors should remove this claim and focus on the novel and exciting DNMT1 biology they have uncovered. This could include some simple follow-up that I felt was lacking in the paper, such as (i) IC50 curves for DAC with various mutants to make quantitative comparisons, (ii) kinetic parameters that alter catalytic activity, (iii) changes in DNA methylation and (iv) validation that the speculated effects on UHRF1 PPIs occur.

Clarity should be given on how LOF mutations can be clustered around the catalytic domain to drive resistance. If gene dosage is speculated, this should be validated with follow-up experiments, or alternatively, a clear reason for why this doesn't occur at other regions of the protein should be given.

---

## [Author Response]

Essential revisions:1) To better support the claim that "activity-based CRISPR scanning" can enrich for resistance mutations in allosteric sites versus orthosteric sites, the reviewers find it would be important to perform a head-to-head comparison with a standard reversible DNMT1 inhibitor. Alternatively, since the reviewers thought that the current screens were of sufficient quality and general interest, the authors could instead elect to tone down the claims about the special utility of activity-based CRISPR scanning in the context of enriching allosteric resistance mechanisms.

We appreciate the reviewers’ enthusiasm with respect to the screening approach and data. To address the reviewers’ concerns, we conducted CRISPR-suppressor scanning on *DNMT1* and *UHRF1* using the non-covalent, reversible DNMT1 inhibitor GSK3484862 (GSKi), which is shown in Figure 1e–h. We observe that the top enriched sgRNA in the GSKi screen targets H1507, which is located near the drug binding site and makes key contacts with the drug (Figure 1e,h, Supplementary Figure 1e). Furthermore, our data indicates that the sgRNA enrichment profiles under decitabine (DAC) versus GSKi treatment are highly distinct (Figure 1g), suggesting that these two classes of activity-based and occupancy-driven inhibitors may exert unique selective pressures that lead to diverse enrichment profiles. Although we consider these data to support the claim that activity-based CRISPR scanning enriches for mutations in allosteric sites versus orthosteric sites, we recognize that allosteric site mutations can also be achieved with non-covalent inhibitors as well. Therefore, we have also modified the text to tone down our claims, suggesting that the use of activitybased inhibitors may exert some bias for the enrichment of allosteric mutations without implying that their enrichment is exclusive to the use of activity-based inhibitors.

2) The Reviewers felt that the manuscript would be strengthened by further characterization of a subset of the resistance mutants to better understand their mechanism of action. Much seems to be inferred from the location of the resistance mutants without experimental follow-up, and these conclusions were viewed as being somewhat speculative at this stage. For instance, the clustering of the apparent LOF mutations around the active site was intriguing, but also perplexing – if reflecting simply a gene dosage effect, why wouldn't such LOF mutations be more broadly distributed throughout the protein? Could a different (and possibly more interesting) mechanism of resistance underpin this cluster of catalytic site mutations? Or, more generally, can the clustering of mutations in the same region of the protein be safely concluded to reflect a shared mechanism for these mutations?

The reviewers make excellent points. We have provided a summary of major additions and changes to the text and experimental data to both points below. A more detailed explanation of these changes can be found in our point-by-point responses to each Reviewer.

With respect to the resistance mechanism of cluster 2 sgRNAs, we recognize that the reasoning underlying our claim of a gene dosage effect may have lacked sufficient explanation and data for clarity. We have included additional text and experiments to further support the idea that cluster 2 sgRNAs may operate through a loss-of-function (LOF)-mediated reduction in gene dosage.

We believe that a major source of ambiguity in the text lies in the definition of “loss-of-function,” which we use in the text as an umbrella term for mutations that are expected to result in a nonfunctional protein product (i.e., frameshift, nonsense, splice site disrupting). All other mutations that preserve the coding frame are designated as “in-frame” with no assumptions on their impact on protein function.

The key insight underlying our gene dosage reduction hypothesis for cluster 2 sgRNAs is that the proportion of in-frame mutations resulting in a null allele depends upon the protein positions that they alter (see Shi, J. *et al.*, *Nat. Biotechnol.* 2015). That is, in-frame mutations located in highly conserved or mutationally constrained regions that are structurally/functionally important are more likely to disrupt protein function than those located in non-essential protein regions (e.g., inter-domain linkers). As a result, sgRNAs targeting essential regions, such as the DNMT1 catalytic domain, are more likely to generate inframe mutations that result in a null allele and are thus “effectively LOF.” This implies that sgRNAs targeting essential protein regions will generate more null alleles than those targeting non-essential regions.

In this context, the observed enrichment of sgRNAs targeting the catalytic domain (e.g., cluster 2) may reflect the ability of these sgRNAs to generate “effectively LOF” in-frame mutations that disrupt DNMT1’s essential function. To support this hypothesis, we performed a time course genotyping analysis on the mutational distribution of 6 individual sgRNAs targeting the N-terminus, RFTS, and catalytic domains (Figure 4f). This experiment is similar in design to those conducted in Shi, J. *et al.*, *Nat. Biotechnol.* 2015. Whereas sgRNAs targeting outside the catalytic domain exhibited increasing frequencies of in-frame mutations over time, consistent with the idea that in-frame mutations in these regions retain function, catalytic-targeting sgRNAs exhibited strong depletion of in-frame mutations over time, supporting the notion that these mutations perturb DNMT1’s essential function and are negatively selected under normal growth conditions. Consequently, the ability of catalytic-targeting sgRNAs to generate greater proportions of null mutations would therefore make them more effective at conferring resistance through reduction of gene dosage than other *DNMT1*-targeting sgRNAs. Altogether, these data support that cluster 2 sgRNAs may operate through a gene dosage reduction effect.

With respect to the characterization of the resistant mutations, we have performed additional experiments and analyses using both recombinant purified protein and cellular assays. A summary of the major additional data are as follows:

1. We performed enzyme activity assays at various temperatures with recombinant DNMT1 protein for WT and mutant DNMT1 constructs, observing that mutant DNMT1 constructs exhibit varying degrees of overactivity relative to WT DNMT1 at different temperatures (Figure 3h, Supplementary Figure 4f). Our data suggest that some gain-of-function DNMT1 mutations may alter the activation energy required to release DNMT1 autoinhibition.

2. We derived clonal cell lines from the activity-based CRISPR scanning screen that contain in-frame mutations in the RFTS and autoinhibitory linker regions (Supplementary Figure 3g). We treated these cell lines and WT K562 cells with varying concentrations of DAC, demonstrating that these clonal lines exhibit partial growth rescue under DAC treatment (Figure 3i). We also include immunoblots demonstrating that these clonal cell lines exhibit DAC-mediated degradation of DNMT1 (Supplementary Figure 3h).

3. To further validate whether the *DNMT1* mutations identified in these clonal cell lines confer DAC resistance, we performed knockdown and overexpression experiments by transducing WT K562 cells with an shRNA expression vector to knock down endogenous DNMT1 and DNMT1 overexpression vectors encoding WT and mutant DNMT1 constructs (Supplementary Figure 3i). We treated these cells with varying concentrations of DAC and demonstrate that overexpression of the mutant DNMT1 constructs also confer partial growth rescue under DAC treatment relative to WT DNMT1 (Figure 3j).

Reviewer #1 (Recommendations for the authors):I think the DNMT1 screening data is sufficiently interesting and rigorous to be appropriate for publication in eLife, but I think one of two changes should be considered. Either a head-to-head comparison should be made to support the claim that this "activity-based CRISPR scanning" yields a preference for allosteric site discovery over "CRISPR suppressor scanning" or the authors should remove this claim and focus on the novel and exciting DNMT1 biology they have uncovered. This could include some simple follow-up that I felt was lacking in the paper, such as (i) IC50 curves for DAC with various mutants to make quantitative comparisons, (ii) kinetic parameters that alter catalytic activity, (iii) changes in DNA methylation and (iv) validation that the speculated effects on UHRF1 PPIs occur.

We really appreciate the very clear and constructive feedback from the reviewer, outlining a clear path to strengthen the manuscript and suitably address their concerns. As we mentioned previously, we have performed CRISPR scanning on *DNMT1* and *UHRF1* using the non-covalent, reversible DNMT1 inhibitor GSK3484862 and performed comparative analyses against our screen with DAC. Our data support our claim that activity-based CRISPR scanning (e.g., DAC) may exert preferential bias for allosteric site enrichment compared to traditional occupancy-based inhibitors (e.g., GSK3484862). We have also toned down the claims regarding activity-based CRISPR scanning to make clear that allosteric site discovery is not exclusive to the use of activity-based inhibitors.

We have also included several further characterization experiments to strengthen the manuscript. We provide enzyme activity data suggesting that hypermorphic *DNMT1* mutations may reduce the activation energy barrier for releasing DNMT1 autoinhibition. We also provide cellular characterization of RFTS and autoinhibitory linker mutations using clonal cell lines harboring endogenous *DNMT1* mutations, as well as knockdown/overexpression experiments, and show that these mutations confer a growth rescue to DAC compared to WT DNMT1.

Clarity should be given on how LOF mutations can be clustered around the catalytic domain to drive resistance. If gene dosage is speculated, this should be validated with follow-up experiments, or alternatively, a clear reason for why this doesn't occur at other regions of the protein should be given.

We have included genotyping time course analysis experiments for catalytic- and non-catalytic-targeting *DNMT1* sgRNAs, demonstrating that catalytic-targeting sgRNAs are more effective at generating “effectively LOF” in-frame mutations. We have also added additional text explaining the reasoning underlying our gene dosage reduction hypothesis that should provide more clarity for the reader.